# Evolution of Information in Interactive Decision Making: A Case Study for Multi-Armed Bandits

**Yuzhou Gu**
New York University
yuzhougu@nyu.edu

**Yanjun Han**
New York University
yanjunhan@nyu.edu

**Jian Qian**
New York University
jianqian@nyu.edu

## Abstract

We study the evolution of information in interactive decision making through the lens of a stochastic multi-armed bandit problem. Focusing on a fundamental example where a unique optimal arm outperforms the rest by a fixed margin, we characterize the optimal success probability and mutual information over time. Our findings reveal distinct growth phases in mutual information—initially linear, transitioning to quadratic, and finally returning to linear—highlighting curious behavioral differences between interactive and non-interactive environments. In particular, we show that optimal success probability and mutual information can be decoupled, where achieving optimal learning does not necessarily require maximizing information gain. These findings shed new light on the intricate interplay between information and learning in interactive decision making.

## 1 Introduction

Consider the following instance of a stochastic multi-armed bandit problem: there are $n$ arms in total, where the optimal arm $a^\star \in [n]$ is uniformly at random, and the reward distribution of arm $i$ is

$$r^i \sim \begin{cases} \text{Ber}\left(\frac{1+\Delta}{2}\right) & \text{if } i = a^\star, \\ \text{Ber}\left(\frac{1-\Delta}{2}\right) & \text{otherwise.} \end{cases} \tag{1}$$

Here $\Delta \in (0,1]$ is a fixed noise parameter. In other words, the best arm $a^\star$ is uniformly better than the rest of the arms by a fixed margin $\Delta$. Readers familiar with the bandit literature shall immediately find that this is the lower bound instance for the multi-armed bandit, where it is well-known (see, e.g., (Lattimore and Szepesvári, 2020, Chapter 15)) that the sample complexity of identifying the best arm is $\Theta\left(\frac{n}{\Delta^2}\right)$. In contrast, it is also a classical result (e.g., via Fano's inequality) that in the non-interactive setting, the sample complexity becomes $\Theta\left(\frac{n \log n}{\Delta^2}\right)$. In other words, the interactive sampling nature of multi-armed bandits offers a $\Theta(\log n)$ gain in the sample complexity compared with the non-interactive sampling.

In this paper, we take a closer look into this seemingly toy example, and investigate how an interactive procedure starts to accumulate information and identify the best arm below the sample complexity, i.e., when $t < \frac{n}{\Delta^2}$. Specifically, denoting by $a_t \in [n]$ the action taken at time $t$ and $\mathcal{H}_t = \sigma(a_1, r_1^{a_1}, \ldots, a_t, r_t^{a_t})$ the available history up to time $t$, we will study the following two quantities:

$$p_t^\star = \sup_{\text{Alg}} \mathbb{P}(a_{t+1} = a^\star), \qquad I_t^\star = \sup_{\text{Alg}} I(a^\star; \mathcal{H}_t). \tag{2}$$

In other words, $p_t^\star$ is the optimal success probability of identifying the best arm $a^\star$ after $t$ rounds, and $I_t^\star$ is the optimal mutual information accumulated through a time horizon of $t$ rounds. Here both supremums are taken over all possible interactive algorithms with the knowledge of $n$ and $\Delta$. In the rest of this paper, we will be interested in the evolution of $p_t^\star$ and $I_t^\star$ as a function of $t$, especially for the curious regime of a small $t$. In fact, before the learner can reliably identify the best arm $a^\star$, the optimal information $I_t^\star$ exhibits a nonlinear accumulation in $t$: for very small $t$ we expect little

difference between interactive and non-interactive settings, so the heuristic from the non-interactive setting would suggest a linear scaling $I_t^\star \asymp \frac{t\Delta^2}{n}$; however, since the optimal bandit algorithm only needs $t \asymp \frac{n}{\Delta^2}$ samples to identify the best arm reliably, we should have $I_t^\star \asymp \log n$ for this choice of $t$, which is $\Theta(\log n)$ larger than the non-interactive heuristic. Again, just like the $\Theta(\log n)$ gain in the sample complexity under the interactive case, even in this toy example, it is unknown when and how interactive learning departs from non-interactive learning and leads to a nonlinear learning curve.

We remark that this stylized toy example is merely used for a case study, and that we do *not* intend to advocate the use of our algorithms in more general settings; they are designed specifically for theoretical analysis. However, in this case study, we find this toy example to be sufficiently illustrative for several interesting phenomena in interactive learning, as well as failures in existing approaches of establishing them:

1. *Understanding the true shape of learning in multi-armed bandits*: The influential work (Garivier et al., 2019) characterizes the "true shape of regret" in bandit problems, where the growth of optimal regret progresses through three regimes: initially linear in time, then squared root in time, and finally logarithmic in time. Building on this, we ask for the "true shape of learning", especially for the initial phase (or the "burn-in" period). Even in the first regime, where the regret grows linearly, the optimal algorithm still engages in nontrivial learning, accumulating information about the environment. Notably, interactive learning plays a pivotal role in this initial phase, enabling a $\Theta(\log n)$ reduction in the sample complexity compared with the non-interactive approaches. Therefore, characterizing the trajectories of $(p_t^\star, I_t^\star)$, even in this toy example, provides deeper insight into the mechanisms of bandit learning beyond regret analysis.

2. *Characterization of mutual information for general interactive decision making*: There have been recent advances on the statistical complexity of general interactive decision making, most notably the DEC (decision-estimation coefficient) framework (Foster et al., 2021, 2022, 2023). One important remaining question in the DEC framework is to close the gap of the so-called "estimation complexity", which precisely corresponds to, in the multi-armed bandits problem, the $\Theta(\log n)$ reduction of the regret. Towards closing this general gap, the recent work (Chen et al., 2024) develops a unified lower bound proposing to keep track of certain notions of information, such as the mutual information; however, this work does not address the problem of how to bound the mutual information in interactive scenarios. This task could be very challenging, as witnessed by another recent work (Rajaraman et al., 2024) which proposes an entirely new line of information-theoretic analysis in the special case of non-linear ridge bandits. Unfortunately, as will be shown later, even in this toy example their tool falls short of giving the right evolution of $I_t^\star$ in some important regimes. Therefore, this work adds new ideas and tools to the literature on bounding the success probability and mutual information in interactive environments.

3. *Exploring the interplay between information and learning*: A more interesting question is whether the high-level proposal of using information to characterize interactive learning in (Chen et al., 2024) could have inherent limitations. In bandit literature, an upper bound of $I_t^\star$ is often translated into an upper bound of $p_t^\star$ (e.g., via Fano's inequality). Conversely, working in the ridge bandit setting inspired by (Lattimore and Hao, 2021; Huang et al., 2021), (Rajaraman et al., 2024) leveraged the reverse direction, critically using an upper bound of $p_t^\star$ to bound the information gain $I_{t+1}^\star - I_t^\star$ in interactive settings. However, it is a priori unclear if some of these links could be strictly loose, where the evolution of $p_t^\star$ (learning) may not always align with the evolution of $I_t^\star$ (information). For instance, the algorithms that achieve optimal learning may not accumulate the largest amount of information. If such discrepancies arise, mutual information alone might not suffice to establish fundamental limits of learning, and new technical tools will be called for. Our multi-armed bandit example is very natural and exactly identifies this important separation.

**Notation and terminology.** Logarithms have base $e$. For two non-negative functions $f$ and $g$, we use $f \lesssim g$ (or $f = O(g)$) to denote that $f \leq Cg$ for a universal constant $C > 0$; $f \gtrsim g$ (or $f = \Omega(g)$) means that $g \lesssim f$; and $f \asymp g$ (or $f = \Theta(g)$) means $f \lesssim g$ and $g \lesssim f$. For probability measures $P$ and $Q$ over the same space, let $\mathrm{TV}(P, Q) = \frac{1}{2}\int |\mathrm{d}P - \mathrm{d}Q|$ be the total variation distance, $\mathsf{H}^2(P, Q) = \int (\sqrt{\mathrm{d}P} - \sqrt{\mathrm{d}Q})^2$ be the squared Hellinger distance, and $\mathrm{KL}(P\|Q) = \int \mathrm{d}P \log \frac{\mathrm{d}P}{\mathrm{d}Q}$ be the Kullback–Leibler divergence. For a joint distribution $P_{XY}$, let $I(X;Y) = \mathrm{KL}\left(P_{XY}\|P_X P_Y\right)$ be the mutual information between $X$ and $Y$. For $x, y \in \mathbb{R}$, let $x \wedge y := \min\{x, y\}$ and $x \vee y := \max\{x, y\}$. Instead of calling upper and lower bounds which may cause confusion, throughout the paper we will use "achievability" to refer to lower bounds of $p_t^\star$ and $I_t^\star$ via constructing explicit algorithms, and "converse" to refer to upper bounds of $p_t^\star$ and $I_t^\star$ that hold for all possible algorithms.

## 1.1 Main results and discussions

Our main result is a complete characterization of the optimal success probability $p_t^\star$ and the optimal mutual information $I_t^\star$.

**Theorem 1.1.** *Assume* $0 < \Delta = 1 - \Omega(1)$. *For* $t \geq 1$, *we have*

$$
p_t^\star \asymp \begin{cases} \frac{1}{n} & \text{if } t\Delta^2 \leq 1 \\ \frac{t\Delta^2}{n} & \text{if } 1 < t\Delta^2 \leq n \ , \\ 1 & \text{if } t\Delta^2 > n \end{cases}
\qquad
I_t^\star \asymp \begin{cases} \frac{t\Delta^2}{n} & \text{if } t\Delta^2 \leq 1 \\ \frac{(t\Delta^2)^2}{n} & \text{if } 1 < t\Delta^2 \leq \log n \\ \frac{t\Delta^2 \log n}{n} & \text{if } \log n < t\Delta^2 \leq n \\ \log n & \text{if } t\Delta^2 > n \end{cases} .
\tag{3}
$$

*Furthermore, all achievablity results can be attained by Algorithm 1 in Section 2.*

**Remark 1.1.** The same characterization also holds for the frequentist counterpart of $p_t^\star$, defined as $p_{t,\mathrm{F}}^\star = \sup_{\mathrm{Alg}} \min_{a^\star \in [n]} \mathbb{P}_{a^\star}(a_{t+1} = a^\star)$. This follows from $p_{t,\mathrm{F}}^\star \leq p_t^\star$, and that the achievability results in Algorithm 1 achieve a frequentist guarantee. However, the definition of the mutual information $I_t^\star$ does not directly extend to the frequentist setting. ◁

**Remark 1.2.** The achievability results for $p_t^\star$ can be obtained using an algorithm which randomly samples $\lceil t\Delta^2 \rceil$ arms and runs a bandit algorithm with optimal regret on them (such as the median elimination algorithm in Even-Dar et al. (2002)). However, it is unclear whether such an algorithm can attain the achievability results for $I_t^\star$. ◁

In comparison, in the non-interactive case, the corresponding quantities are

$$
p_{t,\mathrm{NI}}^\star \asymp \frac{t\Delta^2}{n \log(1 + t\Delta^2)}, \qquad I_{t,\mathrm{NI}}^\star \asymp \frac{t\Delta^2}{n},
\tag{4}
$$

whenever $t \leq \frac{n \log n}{\Delta^2}$. For completeness we include the proof of (4) in Appendix B. Comparing (3) and (4), we see that while $p_{t,\mathrm{NI}}^\star$ is slightly sublinear in $t$ (due to the logarithmic factor on the denominator), $p_t^\star$ becomes precisely linear in $t$ after exiting the easy regime $p_t^\star \asymp \frac{1}{n}$. As will become evident in our algorithm, this difference arises because, in the interactive case, the learner can strategically sample suboptimal arms fewer times.

The evolution of information in the interactive case is more intriguing. While $I_{t,\mathrm{NI}}^\star$ grows linearly in $t$, the growth of $I_t^\star$ features three distinct transitions at $t\Delta^2 \asymp 1, \log n, n$. Intuitively, since $\Theta(1/\Delta^2)$ pulls of an arm estimate its mean reward within accuracy $\Delta$ with a constant probability, we define $m := t\Delta^2 \in [0, n]$ as the "effective" number of arms pulled by the algorithm. Depending on $m$, the evolution of $I_t^\star$ follows four distinct regimes:

1. *First linear regime* $m \leq 1$: In this early stage, the time is too limited to learn even a single arm. The optimal strategy is to query a single arm chosen uniformly at random, making the process non-interactive. Consequently, the growth of $I_t^\star$ is identical in both the interactive and non-interactive cases, exhibiting a linear dependence on $t$.
2. *Quadratic regime* $1 < m \leq \log n$: In this intermediate regime, the time budget suffices to confidently learn one arm but not to achieve $(1 - 1/n)$ confidence. Interaction now plays a crucial role: the learner observes the arm's performance and decides whether to keep pulling it for more confidence or switch to a new arm. The optimal strategy is to stick with the current arm if preliminary estimates suggest it is promising, otherwise switching to explore a different arm. This strategy ensures that the information gain $I_{t+1}^\star - I_t^\star$ is proportional to the probability of pulling the best arm, which increases with $t$ thanks to interaction. As a result, $I_t^\star$ exhibits quadratic growth with $t$.
3. *Second linear regime* $\log n < m \leq n$: In this regime, the best arm can be identified with high confidence if pulled, and pulling it yields diminishing returns in terms of additional information. The total information gain is determined by the probability of identifying the best arm within the time budget, which scales as $m/n$ and again linear in $m$ (and $t$). Compared with the first linear regime, the slope here benefits from an additional $\Theta(\log n)$ factor, for the best arm can now provide $\Theta(\log n)$ bits of information once pulled, again thanks to interaction.
4. *Saturation regime* $m > n$: In the final regime, the learner can reliably identify the best arm, so the quantity $I_t^\star$ saturates at its maximum value $\Theta(\log n)$, the Shannon entropy of $a^\star \sim \mathrm{Unif}([n])$.

Finally, we examine the relationship between learning and information accumulation. A classical inequality between $p_t^\star$ and $I_t^\star$ is the Fano's inequality (Fano, 1968), which in our setting can be expressed as

$$p_t^\star \leq \frac{I_t^\star + h_2(p_t^\star)}{\log n}, \tag{5}$$

where $h_2(p) := p \log \frac{1}{p} + (1-p) \log \frac{1}{1-p}$ is the binary entropy function. Using the upper bound $h_2(p) \leq p \log \frac{1}{p} + p$, this simplifies to:

$$p_t^\star \log \frac{n p_t^\star}{e} \leq I_t^\star. \tag{6}$$

From (3) and (4), it is clear that the relationship (5) (or (6)) is tight in the non-interactive case, and in the interactive regimes where $t\Delta^2 = O(1)$ or $t\Delta^2 = n^{\Omega(1)}$. However, in the intermediate regime of the interactive case where $t\Delta^2 \gg 1$ and $\log(t\Delta^2) = o(\log n)$, Fano's inequality becomes strictly loose. This looseness is not specific to Fano's inequality but applies more broadly to mutual information. Notably, there exists an algorithm in this regime that achieves the optimal success probability $p_t^\star$ while accumulating strictly less information than $I_t^\star$ (see Section 4.2). This indicates that the success probability $p_t^\star$ cannot always be inferred from mutual information alone. This surprising observation suggests that mutual information, while powerful, may be insufficient to fully characterize the fundamental limits of learning in interactive decision making.

The potential looseness of Fano's inequality also introduces new challenges on the technical side. For certain values of $t$, we require tools beyond mutual information to establish tight converse results for $p_t^\star$. Existing approaches fall short, particularly when aiming to prove a small success probability (see Section 4.1). To address these gaps, we propose the following technical innovations for the converse:

1. To upper bound the success probability $p_t^\star$, we devise a reduction scheme which relates $p_t^\star$ for small $t$ to $p_t^\star$ for large $t$, and utilize the classical result $p_t^\star \leq 1 - c$ for some constant value of $c > 0$ when $t = \frac{n}{\Delta^2}$. A noteworthy feature of this reduction is the use of the same algorithm employed in the achievability results as a component of the reduction itself, a novel interplay between these two facets of the analysis.

2. To upper bound the mutual information $I_t^\star$, for $t \leq \frac{\log n}{\Delta^2}$ we critically leverage the upper bound of $p_t^\star$ (established via the aforementioned reduction) to constrain the information gain. This presents an intriguing contrast to Fano's inequality, where $I_t^\star$ is typically used to upper bound $p_t^\star$; here, we reverse the roles and use $p_t^\star$ to bound $I_t^\star$. For large $t$ we use a simple but powerful change-of-divergence technique to obtain the additional $\Theta(\log n)$ factor.

## 1.2 Related work

**Multi-armed bandits.** The multi-armed bandit problem dates back to (Robbins, 1952; Lai and Robbins, 1985). Early algorithms like UCB and EXP3 achieved an $\widetilde{O}(\sqrt{nT})$ minimax regret bound, which is tight up to logarithmic factors (Auer et al., 1995, 2002a,b). (Audibert and Bubeck, 2009) first removed this extra logarithmic factor—the key difference between interactive and non-interactive settings—via a new potential (INF policy) or an optimistic upper confidence bound (MOSS algorithm). Extensions of these algorithms, such as the anytime variant (Degenne and Perchet, 2016) and best-of-both-worlds guarantees (Zimmert and Seldin, 2019), are also available in the literature. However, a deeper understanding of the trajectories of these algorithms in the initial learning period is still missing, and the tight separation between non-interactive and interactive algorithms largely remains a mystery for general decision making after a rich line of DEC developments (Foster et al., 2021, 2022, 2023; Chen et al., 2024). A similar mystery holds for the mutual information: for example, while information-directed sampling (Russo and Van Roy, 2018) seeks to maximize the information gain in the initial steps, its analysis critically relies on the information ratio (Russo and Van Roy, 2016) and does not provide insights into the evolution of information.

**Best arm identification.** Our problem formulation in (1), modulo the specific prior $a^\star \sim \text{Unif}([n])$, aligns with best arm identification in multi-armed bandits. Algorithms based on various principles such as the frequentist method of UCB (Bubeck et al., 2011) and arm elimination (Audibert et al., 2010) or the Bayesian method of knowledge gradient (Frazier and Powell, 2008) and Thompson sampling (Russo, 2016), have been proposed for best arm identification. In particular, the asymptotic complexity in the fixed confidence setting has been completely characterized in (Garivier and

Kaufmann, 2016), and progress has also been made on the fixed budget setting (Carpentier and Locatelli, 2016; Kato et al., 2022; Komiyama et al., 2022). The stopping rule used in our Algorithm 1 is inspired by this body of work and relies on the sequential probability ratio test dating back to Wald (Wald, 1945). Under our problem formulation, it is also known that the "median elimination algorithm" in (Even-Dar et al., 2002; Mannor and Tsitsiklis, 2004) achieves the optimal sample complexity without the $\log n$ factor. However, most existing guarantees for best arm identification are inapplicable for small time horizons $t$ or when the error probability is $1 - o(1)$. For example, although (Audibert et al., 2010) established a lower bound of $\exp(-(c + o(1))t\Delta^2/n)$ on the error probability specialized to our setting, the $o(1)$ factor becomes negligible only when $t \geq n^2/\Delta^2$. As another example, the error probability upper bound $\exp(-(c + o(1))t/(H \log n))$ in the fixed budget setting of (Komiyama et al., 2022), with $H \asymp \frac{n}{\Delta^2}$ in our setting, has an extra $O(\log n)$ factor and requires a large $t \gg n^2$. Similar requirements for large $t$ are also present in the results of (Katz-Samuels and Jamieson, 2020; Zhao et al., 2023; Carpentier and Locatelli, 2016; Karnin et al., 2013; Atsidakou et al., 2022). In contrast, our work establishes the tight probability of success for small $t \leq \frac{n}{\Delta^2}$ as well, and identifies curious phase transitions in the mutual information behind the large error probabilities.

**Feedback communication and noisy computation.** The objectives of minimizing error probability and maximizing mutual information align closely with concepts from feedback communication, a classical topic in information theory (Burnašev, 1980; Tatikonda and Mitter, 2008). For instance, (Burnašev, 1980) established upper bounds on mutual information that reveal two distinct phases in interactive environments, which parallel the "burn-in phase" and "learning phase" in our problem. Although interactive decision making operates under a more constrained model than communication systems—learners can only pull one of $n$ arms rather than utilize an arbitrary encoder—the perspective in (Burnashev, 1976) has proven valuable in addressing recent noisy computation challenges, such as noisy sorting (Wang et al., 2024), particularly for deriving converse results. However, unlike in feedback communication, our multi-armed bandit problem does not always exhibit linear growth in mutual information during the "burn-in phase".

Our problem can also be framed as a novel noisy computation task, where the learner would like to locate the maximum of a random permutation of $((1 + \Delta)/2, (1 - \Delta)/2, \ldots, (1 - \Delta)/2)$ through noisy queries. Recent years have seen a resurgence of interest in such problems, revisiting classical questions in noisy sorting (Wang et al., 2024; Gu and Xu, 2023) and the noisy computation of threshold (Wang et al., 2025; Gu et al., 2025), MAX, and OR functions (Zhu et al., 2024). These works often leverage a powerful converse technique from (Feige et al., 1994), which reduces interactive environments to a two-phase process comprising a non-interactive phase and an interactive phase that returns the clean output. We will show in Section 4.1 that this approach does not directly succeed in our problem. Our converse results on success probability are derived using a distinct reduction method.

**Converse techniques.** Beyond the techniques in (Burnašev, 1980; Feige et al., 1994) discussed earlier, we review additional methods used to establish converse results in the statistics and bandit literature. The most common approach for proving regret lower bounds in multi-armed bandits relies on a change-of-measure argument (Lai and Robbins, 1985) (see also (Lattimore and Szepesvári, 2020, Chapter 15.2) for minimax lower bounds and (Simchowitz et al., 2017) for more advanced treatments). However, as these methods are special cases of Le Cam's two-point method, the resulting lower bound on error probability cannot exceed $1/2$ (achievable by a random coin flip). Even when generalized to test multiple hypotheses, as we show in Section 4.1, this approach still yields a weaker converse result $p_t^\star = O(1/n + (\sqrt{t/n} \wedge (t/n))\Delta)$.

Controlling mutual information can overcome the limitations of the two-point method. Despite that early works (Agarwal et al., 2012; Raginsky and Rakhlin, 2011a,b) showed that the amount information acquired by an *interactive* algorithm could be harder to quantify, this approach has been revisited recently in the interactive framework (Rajaraman et al., 2024; Chen et al., 2024). For instance, (Chen et al., 2024) extended the idea of (Chen et al., 2016) to develop an algorithmic version of Fano's inequality for interactive settings, but did not address the critical problem of bounding mutual information. This challenging task was tackled in (Rajaraman et al., 2024) in the context of ridge bandits using an induction argument that required the success probability at each step to be exponentially small, allowing the application of a union bound. However, this approach fails for multi-armed bandits, as even a random guess achieves a success probability of $\Omega(1/n)$ at each step, which is not small enough to apply union bound. This is the high-level reason why such an

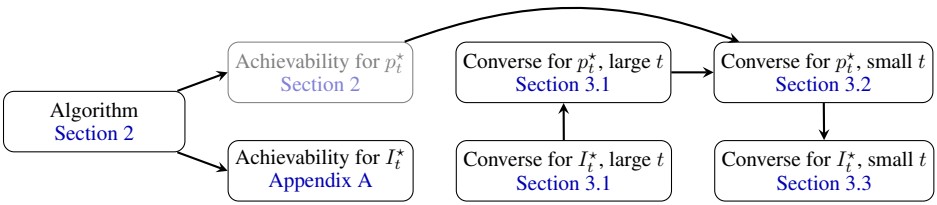

Figure 1: Dependency between different parts of our proof of Theorem 1.1. We make one node opaque to highlight that achievability results for $p_t^\star$ are also available in the literature (cf. Remark 1.2).

argument can only be applied for small $t \leq \frac{\log n}{\Delta^2}$; for larger $t$, a different technique—based on a *change-of-divergence* argument—provides the correct upper bound on mutual information. While conceptually simple, this method offers the first known proof (to the authors' knowledge) of lower bounds in multi-armed bandits using mutual information and Fano's inequality.

### 1.3 Organization

The rest of the paper is organized as follows. In Section 2, based on the sequential probability ratio test, we introduce a simple algorithm (Algorithm 1) for identifying the best arm $a^\star$. Based on the theory of biased random walks, we show that this algorithm achieves the claimed success probability (in Section 2) and mutual information (in Appendix A). Section 3 establishes the converse results for $p_t^\star$ and $I_t^\star$, and the proof distinguishes into the cases of large $t$ and small $t$. For large $t$, we directly establish an upper bound of $I_t^\star$ and apply Fano's inequality to upper bound $p_t^\star$. The converse analysis of small $t$ is more involved, where the upper bound of $p_t^\star$ is proven via a reduction to the case of large $t$, with the help of the same algorithm in Section 2. Furthermore, this upper bound of $p_t^\star$ is crucially used in an information-theoretic argument to upper bound $I_t^\star$. Dependency between different parts of our proof of Theorem 1.1 is shown in Figure 1.

We provide some discussions in Section 4. Specifically, we establish the suboptimality of several existing approaches for the converse in Section 4.1, and prove the separation between learning and information gain in Section 4.2.

## 2 Achievability

In this section we show a simple algorithm can strategically sample suboptimal arms fewer times and achieve the success probability lower bounds in Theorem 1.1. This algorithm is based on the sequential probability ratio test for $H_0 : r \sim \text{Ber}\left(\frac{1-\Delta}{2}\right)$ against $H_1 : r \sim \text{Ber}\left(\frac{1+\Delta}{2}\right)$, described in Algorithm 1.

---

**Algorithm 1** SEQUENTIALPROBABILITYRATIOTEST$(A, t, \Delta)$

---

1: **input:** action set $A$, number of rounds $t$, noise parameter $\Delta$
2: **output:** an estimate of the best arm $\widehat{a} \in A$
3: Permute $A$ uniformly at random. Relabel elements of $A$ as $1, \ldots, n$ where $n = |A|$.
4: $\theta \leftarrow -1/\Delta, s \leftarrow 0$
5: **for** $i = 1$ to $n$ **do**
6: $\quad$ $X^i \leftarrow 0$
7: $\quad$ **while** true **do**
8: $\quad\quad$ **if** $s = t$ **then return** $\widehat{a} = i$
9: $\quad\quad$ Pull action $i$ and receive reward $r_t^i \in \{0, 1\}$
10: $\quad\quad$ $X^i \leftarrow X^i + 2r_t^i - 1, s \leftarrow s + 1$ $\qquad\qquad$ ▷ Random walk for $X^i$
11: $\quad\quad$ **if** $X^i \leq \theta$ **then break**
12: **return** $\widehat{a} = 1$

---

Under Bernoulli rewards, the sequential probability ratio test is a biased random walk for each arm, with bias $\Delta$ for the best arm and bias $-\Delta$ for all suboptimal arms. Algorithm 1 eliminates an arm once its walk drops below the threshold $\theta = -1/\Delta$, and keeps pulling the arm otherwise. The key property of biased random walks is that, it only takes $O(1/\Delta^2)$ steps in expectation for a suboptimal arm to reach the threshold $\theta$, while with $\Omega(1)$ probability the best arm never hits $\theta$. This is a consequence of the following standard results on stopping times of a biased random walk (e.g., (Feller, 1970)).

**Lemma 2.1** (Stopping time of biased random walk). *Let $\theta < 0$ and $0 < \Delta < 1$. Let $(X_t)_{t\geq 0}$ be a random walk starting from $0$ with i.i.d. steps drawn from some distribution $D$, which is either $D_- = \frac{1-\Delta}{2}\delta_1 + \frac{1+\Delta}{2}\delta_{-1}$ or $D_+ = \frac{1+\Delta}{2}\delta_1 + \frac{1-\Delta}{2}\delta_{-1}$. Let $T \in \mathbb{N} \cup \{+\infty\}$ be the stopping time for $X_T \leq \theta$. Then the following statements hold:*

(a) *For the downward random walk $D = D_-$, $\mathbb{E}T \leq -\frac{\theta}{\Delta}$;*

(b) *For the upward random walk $D = D_+$, $\mathbb{P}(T < \infty) \leq \left(\frac{1-\Delta}{1+\Delta}\right)^{-\theta} \leq \exp(2\Delta\theta)$.*

Based on Lemma 2.1, we prove that Algorithm 1 achieves the success probability lower bounds in Theorem 1.1. We restate the result here for convenience.

**Theorem 2.1** (Achievability for success probability). *Algorithm 1 achieves the success probability bounds in Theorem 1.1.*

In the remainder of this section, we prove Theorem 2.1 for $t \leq \frac{n}{\Delta^2}$, as a larger $t$ only makes learning easier. First, we restate the algorithm:

1. Permute the arms uniformly at random.
2. For each arm $i$, the random walk $(X_j^i)_{j\geq 0}$ starts at $0$ with steps from $D_+$ if $i$ is the best arm, and from $D_-$ otherwise. All walks are independent.
3. For each $i \in [n]$, let $T_i$ be the first time that $X_{T_i}^i \leq \theta = -1/\Delta$. Return arm $i$ where $i$ is the smallest index such that $\sum_{k=1}^{i} T_k > t$. If no such $i$ exists (i.e. all arms are eliminated), return arm 1.

Now let $m = \lceil 0.1t\Delta^2 \rceil \leq n$ and define three events: let $\mathcal{E}_1$ be the event that the best arm is among the first $m$ arms (after the random permutation); let $\mathcal{E}_2$ be the event that $\sum_{1\leq i\leq m, i\neq i^\star} T_i \leq t$, where $i^\star$ is the optimal arm after the random permutation; let $\mathcal{E}_3$ be the event that $T_{i^\star} = \infty$.

Clearly, when $\mathcal{E}_1 \cap \mathcal{E}_2 \cap \mathcal{E}_3$ holds, Algorithm 1 outputs the best arm. It remains to lower bound the success probability $\mathbb{P}(\mathcal{E}_1 \cap \mathcal{E}_2 \cap \mathcal{E}_3)$. Clearly $\mathbb{P}(\mathcal{E}_1) = \frac{m}{n}$, and $\mathbb{P}(\mathcal{E}_3^c | \mathcal{E}_1) \leq \exp(2\Delta\theta) = e^{-2} < 1/5$ by Lemma 2.1. In addition, by Markov's inequality, $\mathbb{P}(\mathcal{E}_2^c | \mathcal{E}_1) \leq \frac{1}{t}\mathbb{E}\left[\sum_{1\leq i\leq m, i\neq i^\star} T_i\right] \leq \frac{m-1}{t\Delta^2} \leq 0.1$ by Lemma 2.1 and the definition of $m$. Therefore,

$$\mathbb{P}(\mathcal{E}_1 \cap \mathcal{E}_2 \cap \mathcal{E}_3) \geq \mathbb{P}(\mathcal{E}_1)\left(1 - \mathbb{P}(\mathcal{E}_2^c | \mathcal{E}_1) - \mathbb{P}(\mathcal{E}_3^c | \mathcal{E}_1)\right) \geq \frac{0.7m}{n} = \Omega\left(\frac{1 + t\Delta^2}{n}\right),$$

which is our target. We defer the proofs of achievability results for mutual information to Appendix A.

## 3 Converse

In this section, we prove converse results on $p_t^\star$ and $I_t^\star$ as illustrated in Figure 1.

### 3.1 Converse for large $t$

This section establishes the following converse results for $I_t^\star$ and $p_t^\star$ for large $t$.

**Theorem 3.1** (Converse for large $t$). *The following statements hold under the setting of Theorem 1.1:*

(a) *For any $t \geq 0$, we have $I_t^\star \lesssim \frac{t\Delta^2 \log n}{n} \wedge \log n$.*

(b) *For any $t \geq \frac{n^{\Omega(1)}}{\Delta^2}$, we have $p_t^\star \lesssim \frac{t\Delta^2}{n} \wedge 1$.*

The remainder of this section is devoted to the proof of Theorem 3.1.

**Mutual information.** We apply a "change-of-divergence" argument to prove the converse for $I_t^\star$. Recall that $\mathcal{H}_t$ denotes the history up to time $t$ and $a^\star$ denotes the optimal arm. For any fixed algorithm, we have

$$I(a^\star; \mathcal{H}_t) = \mathbb{E}_{a^\star}\left[\text{KL}\left(P_{\mathcal{H}_t|a^\star} \| P_{\mathcal{H}_t}\right)\right] \overset{(a)}{\leq} (2 + \log n)\mathbb{E}_{a^\star}\left[\text{H}^2\left(P_{\mathcal{H}_t|a^\star}, P_{\mathcal{H}_t}\right)\right]$$

$$\overset{(b)}{\leq} 2(2 + \log n)\inf_{\overline{P}_{\mathcal{H}_t}}\left(\mathbb{E}_{a^\star}\left[\text{H}^2\left(P_{\mathcal{H}_t|a^\star}, \overline{P}_{\mathcal{H}_t}\right)\right] + \text{H}^2\left(\overline{P}_{\mathcal{H}_t}, P_{\mathcal{H}_t}\right)\right)$$

$$\overset{(c)}{\leq} 4(2 + \log n)\inf_{\overline{P}_{\mathcal{H}_t}}\mathbb{E}_{a^\star}\left[\text{H}^2\left(P_{\mathcal{H}_t|a^\star}, \overline{P}_{\mathcal{H}_t}\right)\right] \overset{(d)}{\leq} 4(2 + \log n)\inf_{\overline{P}_{\mathcal{H}_t}}\mathbb{E}_{a^\star}\left[\text{KL}\left(\overline{P}_{\mathcal{H}_t} \| P_{\mathcal{H}_t|a^\star}\right)\right]$$

---

**Algorithm 2** BOOSTING$(A, t, \Delta, m, \mathcal{A})$

---

1: **input:** action set $A$, number of rounds $t$, noise parameter $\Delta$, boosting parameter $m$, an algorithm $\mathcal{A}$ for best arm identification with time budget $t$
2: **output:** an estimate of the best arm $\widehat{a} \in A$
3: $B \leftarrow \emptyset$
4: **for** $i$ from 1 to $m$ **do**
5:      Permute $A$ uniformly at random
6:      Let $a_i \leftarrow \mathcal{A}(A, t, \Delta)$ be the best arm estimate returned by algorithm $\mathcal{A}$ in $t$ rounds
7:      $B \leftarrow B \cup \{a_i\}$
8: Remove duplicate elements from $B$ and let $m'$ be the remaining size
9: **return** $\widehat{a} = $ SEQUENTIALPROBABILITYRATIOTEST$(B, m'/\Delta^2, \Delta)$      ▷ Algorithm 1

---

where (a) uses (Yang and Barron, 1998, Lemma 4) and notes that thanks to $a^\star \sim \mathrm{Unif}([n])$, the density ratio of the two arguments in the KL divergence is upper bounded by $n$ almost surely, steps (b) and (c) use the triangle inequality and convexity of the Hellinger distance, respectively, and (d) follows from $\mathsf{H}^2(P, Q) \leq \mathrm{KL}(P \| Q)$.

Now let $\overline{P}_{\mathcal{H}_t}$ be the dummy model where all arms have reward distribution $\mathrm{Ber}\left(\frac{1-\Delta}{2}\right)$, and $\overline{\mathbb{E}}$ be the expectation taken with respect to $\overline{P}_{\mathcal{H}_t}$. By the chain rule of the KL-divergence,

$$
\mathbb{E}_{a^\star}\left[\mathrm{KL}\left(\overline{P}_{\mathcal{H}_t} \| P_{\mathcal{H}_t | a^\star}\right)\right] = \mathbb{E}_{a^\star}\left[\sum_{s=1}^{t} \overline{\mathbb{E}}\left[\mathrm{KL}\left(\mathrm{Ber}\left(\frac{1+\Delta}{2}\right) \middle\| \mathrm{Ber}\left(\frac{1-\Delta}{2}\right)\right)\mathbb{1}(a_s = a^\star)\right]\right]
$$

$$
\lesssim \Delta^2 \mathbb{E}_{a^\star}\left[\sum_{s=1}^{t} \overline{\mathbb{E}}[\mathbb{1}(a_s = a^\star)]\right] = \frac{t\Delta^2}{n},
$$

where the first inequality follows by the assumption that $\Delta = 1 - \Omega(1)$ and the second identity follows by the symmetry between all arms for both $\overline{P}$ and $a^\star$. Combining all above, we get $I_t^\star \lesssim \frac{t\Delta^2 \log n}{n}$. The other upper bound $I_t^\star \leq H(a^\star) = \log n$ is trivial.

**Success probability.** Suppose that $t \geq \frac{n^c}{\Delta^2}$ for some constant $c > 0$. By the upper bound of $I_t^\star$ and Fano's inequality (6), we have $p_t^\star \log \frac{np_t^\star}{e} \leq I_t^\star \lesssim \frac{t\Delta^2 \log n}{n}$. If $p_t^\star \geq \frac{t\Delta^2}{n}$, then $p_t^\star \log \frac{np_t^\star}{e} \geq p_t^\star \log \frac{n^c}{e} \gtrsim cp_t^\star \log n$. Combining both inequalities we conclude that $p_t^\star \lesssim \frac{t\Delta^2}{n}$, and $p_t^\star \leq 1$ trivially.

### 3.2 Success probability for small $t$

This section establishes the converse results for $p_t^\star$ in the entire range of $t \leq n/\Delta^2$.

**Theorem 3.2** (Success probability converse for small $t$). *Under the setting of Theorem 1.1, we have $p_t^\star \lesssim \frac{1+t\Delta^2}{n}$ for $t \leq \frac{n}{\Delta^2}$.*

The proof of Theorem 3.2 is via a reduction from the success probability lower bound for large $t$ and the following boosting argument.

**Proposition 3.1.** *For any integers $t, m \geq 0$ it holds that $p_{mt+m/\Delta^2}^\star \gtrsim 1 - (1 - p_t^\star)^m$.*

**Proof.** Suppose $\mathcal{A}$ is an algorithm that achieves the optimal success probability $p_t^\star$ using $t$ pulls. Let $\mathcal{A}_{\mathrm{boost}}$ be the boosting algorithm given in Algorithm 2, which runs algorithm $\mathcal{A}$ $m$ times to obtain a candidate action set $B$ of size $m' \leq m$, and then runs Algorithm 1 on the action set $B$. We establish Proposition 3.1 by showing that $\mathcal{A}_{\mathrm{boost}}$ always uses at most $mt + m/\Delta^2$ pulls and achieves success probability $\Omega\left(1 - (1 - p_t^\star)^m\right)$.

**Number of pulls.** Pulls are used only in Lines 6 and 9. Line 6 is executed $m$ times and each time uses $t$ pulls. Line 9 uses $m'/\Delta^2$ pulls with $m' \leq m$. Therefore the total number of pulls is at most $mt + m/\Delta^2$.

**Success probability.** Because we permute the arms uniformly at random before each call in Line 6, the event that each call succeeds (i.e. outputs the best arm $a^\star$) are independent. Let $\mathcal{E}$ be the event that the optimal arm is in $B$, then $\mathbb{P}[\mathcal{E}] \geq 1 - (1 - p_t^\star)^m$. Conditioned on $\mathcal{E}$, there is a unique optimal arm in $B$. By Theorem 2.1, Line 9 succeeds with probability $\Omega(1)$. Therefore the overall success

probability of $\mathcal{A}_{\text{boost}}$ is $\Omega\left(1 - (1 - p_t^\star)^m\right)$. $\qquad\square$

We are now ready to prove Theorem 3.2. By Proposition 3.1, there exists $c_1 > 0$ such that $p_{mt+m/\Delta^2}^\star \geq c_1(1 - (1 - p_t^\star)^m)$. By Theorem 3.1(b), for any $c_2 > 0$, there exists $c_3 > 0$ with $p_{c_3n/\Delta^2}^\star \leq c_2$. Take $c_2 = c_1/2$ and choose $c_3$ accordingly.

Let $m = \lfloor c_3 n/(t\Delta^2 + 1)\rfloor$, so that $mt + m/\Delta^2 \leq c_3 n/\Delta^2$. By the previous paragraph, we have $\frac{c_1}{2} \geq p_{c_3n/\Delta^2}^\star \geq c_1\left(1 - (1 - p_t^\star)^m\right) \geq c_1\left(1 - e^{-mp_t^\star}\right)$. Solving this inequality gives that $p_t^\star \leq \frac{\log 2}{m} \lesssim \frac{t\Delta^2+1}{n}$, establishing Theorem 3.2.

### 3.3 Mutual information for small $t$

This section establishes the converse for $I_t^\star$ when $t \leq \frac{\log n}{\Delta^2}$. Note that when $t > \frac{\log n}{\Delta^2}$, the converse result in Theorem 3.1 is already tight.

**Theorem 3.3** (Mutual information converse for small $t$). *The following statements hold under the setting of Theorem 1.1:*

*(a) For $t \leq \frac{1}{\Delta^2}$, we have $I_t^\star \lesssim \frac{t\Delta^2}{n}$.*
*(b) For $\frac{1}{\Delta^2} < t \leq \frac{\log n}{\Delta^2}$, we have $I_t^\star \lesssim \frac{(t\Delta^2)^2}{n}$.*

The proof of Theorem 3.3 follows from Lemma 3.1 and Theorem 3.2 by a simple calculation.

**Lemma 3.1.** *For any $t$, it holds that $I_t^\star \lesssim \Delta^2 \sum_{s=0}^{t-1} p_s^\star$.*

**Proof.** Given any learning algorithm, let $I_s = I(a^\star; \mathcal{H}_s)$ be the mutual information accumulated by this algorithm until time $s$. In the sequel we upper bound the information gain $I_s - I_{s-1}$ using the upper bounds of $p_s^\star$ in Theorem 3.2.

By the chain rule of the mutual information, we have $I_s - I_{s-1} = I(a^\star; r_s^{a_s}|\mathcal{H}_{s-1}, a_s)$. By the variational representation (e.g., (Polyanskiy and Wu, 2025, Theorem 4.1)) of the mutual information $I(X; Y|Z) = \min_{Q_{Y|Z}} \mathbb{E}_{X,Z}\left[\text{KL}\left(P_{Y|X,Z}\|Q_{Y|Z}\right)\right]$, we have

$$
\begin{aligned}
I(a^\star; r_s^{a_s}|\mathcal{H}_{s-1}, a_s) &\leq \mathbb{E}_{(a^\star, a_s, \mathcal{H}_{s-1})}\left[\text{KL}\left(P_{r_s^{a_s}|a^\star, a_s, \mathcal{H}_{s-1}}\,\Big\|\,\text{Ber}\left(\frac{1-\Delta}{2}\right)\right)\right] \\
&= \mathbb{E}_{(a^\star, a_s, \mathcal{H}_{s-1})}\left[\text{KL}\left(\text{Ber}\left(\frac{1-\Delta}{2} + \Delta\mathbb{1}(a_s = a^\star)\right)\,\Big\|\,\text{Ber}\left(\frac{1-\Delta}{2}\right)\right)\right] \\
&\lesssim \mathbb{E}_{(a^\star, a_s, \mathcal{H}_{s-1})}\left[\Delta^2\mathbb{1}(a_s = a^\star)\right] \leq \Delta^2 p_{s-1}^\star.
\end{aligned}
$$

Summing over $s = 1, \ldots, t$ completes the proof. $\qquad\square$

## 4 Discussions

### 4.1 Failure of existing approaches for converse

We comment on the failure of existing approaches in fully establishing the converse results in Theorem 1.1. The standard bandit lower bound (see, e.g. (Lattimore and Szepesvári, 2020, Chapter 15.2)) uses binary hypothesis testing arguments, and a generalization to the uniform prior $a^\star \sim \text{Unif}([n])$ (a variant of (Gao et al., 2019, Lemma 3)) reads as

$$
\mathbb{P}(a_{t+1} = a^\star) \leq \frac{1}{n} + \inf_{\mathbb{P}_0} \frac{1}{n}\sum_{i=1}^{n} \text{TV}(\mathbb{P}_0, \mathbb{P}_i). \tag{7}
$$

Here $\mathbb{P}_i$ denotes the distribution of $\mathcal{H}_t$ under $a^\star = i$, and $\mathbb{P}_0$ is any reference distribution. By choosing $\mathbb{P}_0$ to be the case where all reward distributions are $\text{Ber}\left(\frac{1-\Delta}{2}\right)$, it is easy to see that (7) gives the tight bound $p_t^\star \leq n^{-1}$ for $t = 0$ and $p_t^\star = 1 - \Omega(1)$ for $t \asymp \frac{n}{\Delta^2}$. However, for intermediate values of $t$, (7) only gives the following lower bound, which does not exhibit the correct scaling on $\Delta$.

**Proposition 4.1.** *For any $t \leq \frac{n}{\Delta^2}$, there exists a learner such that $\inf_{\mathbb{P}_0} \frac{1}{n}\sum_{i=1}^{n} \text{TV}(\mathbb{P}_0, \mathbb{P}_i) = \Omega\left(\left(\frac{t}{n} \wedge \sqrt{\frac{t}{n}}\right)\Delta\right)$.*

Next, we consider the two-phase approach in (Feige et al., 1994), whose failure is more delicate. Specialized to our problem, the idea in (Feige et al., 1994) is to consider a stronger model with two phases: 1) in the non-interactive phase, each arm is pulled $1/\Delta^2$ times; 2) in the interactive case, the learner can query $m = \lceil t\Delta^2 \rceil$ arms based on the outcome from the non-interactive phase, and an oracle gives *clean answers* to the learner about the true mean rewards of the queried arms. Clearly this model can simulate our original model, so a converse on this stronger model taking the form $p_t^\star \lesssim \frac{m}{n}$ (i.e. the second phase is still a "random guess") would establish the converse in the original model. However, the following result shows that the learner can perform strictly better under the new model.

**Proposition 4.2.** *For $t = o(\frac{n}{\Delta^2})$ and $\Delta = o(\frac{1}{\sqrt{\log n}})$, there exists a learner under the two-phase model which achieves a success probability $\gtrsim \frac{t\Delta^2}{n} \exp\left(\Omega\left(\sqrt{\log \frac{n}{t\Delta^2}}\right)\right) = \omega\left(\frac{t\Delta^2}{n}\right)$.*

Finally we discuss the inductive proof of the converse result of (Rajaraman et al., 2024), which inspires our argument in Lemma 3.1. Its failure is straightforward: each inductive step of (Rajaraman et al., 2024) derives an upper bound of $p_t^\star$ from the current upper bound of $I_t^\star$, which in view of the following section is inherently loose for $t\Delta^2 \in \omega(1) \cap n^{o(1)}$.

### 4.2 Optimal algorithm with suboptimal mutual information

In the introduction, by comparing Theorem 1.1 with (5) and (6), we see that Fano's inequality does not give a tight relationship between $p_t^\star$ and $I_t^\star$ when $t\Delta^2 \in \omega(1) \cap n^{o(1)}$. One may wonder if some relationship between $p_t$ and $I_t$, other than Fano's inequality, turns out to be tight for any learning algorithm. The answer turns out to be negative, as shown by the following result where an optimal algorithm may achieve suboptimal mutual information.

**Proposition 4.3.** *There exists a learner achieving the optimal success probability $\Theta(p_t^\star)$ for $t \leq \frac{n}{\Delta^2}$, but a suboptimal mutual information $O\left(\frac{t\Delta^2 \log(t\Delta^2)}{n}\right) = o(I_t^\star)$ if $t\Delta^2 \in \omega(1) \cap n^{o(1)}$.*

Proposition 4.3 implies that a generic relationship $p_t \leq f(I_t)$ that holds for any algorithm cannot be used to establish the tight converse for the success probability. The new algorithm is described in Algorithm 3 in the Appendix; the main difference from Algorithm 1 is an upper threshold $\theta_r > 0$ for the random walks such that the best arm is pulled for fewer times (i.e. early stopping), ensuring the same success probability but providing less information.

### 4.3 Fixed time budget vs stopping time

Another interesting question is how our results change when the fixed time budget $t$ is relaxed to a stopping time with expectation at most $t$. Let $p_{t,\mathrm{E}}^\star$ and $I_{t,\mathrm{E}}^\star$ be the corresponding quantities when the algorithm can stop at such a stopping time, clearly $p_{t,\mathrm{E}}^\star \geq p_t^\star$ and $I_{t,\mathrm{E}}^\star \geq I_t^\star$. The following result gives a tight characterization for $p_{t,\mathrm{E}}^\star$ and an achievability result for $I_{t,\mathrm{E}}^\star$.

**Proposition 4.4.** *For $t \leq \frac{n}{\Delta^2}$, it holds that $p_{t,\mathrm{E}}^\star \asymp \frac{1+t\Delta^2}{n}$ and $I_{t,\mathrm{E}}^\star \gtrsim \frac{t\Delta^2 \log n}{n}$.*

Comparing Proposition 4.4 with Theorem 1.1, one observes that the elbows for the optimal mutual information evaporate upon allowing a random stopping time. Intuitively, by randomization a learner can achieve the upper convex envelope of $I_t^\star$, so that $I_{t,\mathrm{E}}^\star$ exhibits a fast linear growth even at the very beginning. We conjecture that the achievability result for $I_{t,\mathrm{E}}^\star$ in Proposition 4.4 is tight; see Appendix C.5 for more discussions.

## Acknowledgments and Disclosure of Funding

YH is supported in part by an AI for Math grant from Renaissance Philanthropy. JQ acknowledges support from ARO through award W911NF-21-1-0328, as well as the Simons Foundation and the NSF through awards DMS-2031883 and PHY-2019786.

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

# A  Achievability for Mutual Information

In this section we prove that Algorithm 1 achieves the mutual information lower bounds in Theorem 1.1. We restate the result here for convenience.

**Theorem A.1** (Achievability for mutual information). *Under the setting of Theorem 1.1, for $t \leq \frac{n}{\Delta^2}$, Algorithm 1 achieves mutual information*

*(a)* $\Omega\left(\frac{t\Delta^2}{n}\right)$ *when* $0 \leq t \leq \frac{1}{\Delta^2}$;

*(b)* $\Omega\left(\frac{t^2\Delta^4}{n}\right)$ *when* $\frac{1}{\Delta^2} < t \leq \frac{\log n}{\Delta^2}$;

*(c)* $\Omega\left(\frac{t\Delta^2 \log n}{n}\right)$ *when* $\frac{\log n}{\Delta^2} < t \leq \frac{n}{\Delta^2}$.

Because the optimal arm is chosen uniformly at random, the random permutation step in Algorithm 1 does not affect the mutual information achieved by the algorithm, and we remove it in the proof.

## A.1  Proof of (a)

By chain rule of mutual information,

$$I(a^\star; \mathcal{H}_t) = \sum_{0 \leq s \leq t-1} I\left(a^\star; r_{s+1}^{a_{s+1}} | \mathcal{H}_s, a_{s+1}\right). \tag{8}$$

We prove that for $s \leq \frac{1}{\Delta^2}$, we have $I(a^\star; r_{s+1}^{a_{s+1}} | \mathcal{H}_s, a_{s+1}) = \Omega\left(\frac{\Delta^2}{n}\right)$, with a hidden constant independent of $s$. For a fixed $s \leq \frac{1}{\Delta^2}$, define $\mathcal{E}$ as the following event:

1. The first $s$ pulls are to the same arm (i.e., $a_1 = \cdots = a_s = 1$).

2. $-\frac{1}{\Delta} < X_s^1 \leq s\Delta + \frac{2}{\Delta}$, where $X_s^1$ denotes the value of $X^1$ at time $s$.

We prove that

$$\mathbb{P}(\mathcal{E}) = \Omega(1), \tag{9}$$

$$I\left(a^\star; r_{s+1}^{a_{s+1}} | \mathcal{H}_s, a_{s+1}, \mathcal{E}\right) = \Omega\left(\frac{\Delta^2}{n}\right). \tag{10}$$

Since the event $\mathcal{E}$ can be determined by $\mathcal{H}_s$, (9) and (10) imply that $I\left(a^\star; r_{s+1}^{a_{s+1}} | \mathcal{H}_s, a_{s+1}\right) = \Omega\left(\frac{\Delta^2}{n}\right)$, which implies the desired lower bound by (8).

**Proof of (9).** Let $(Z_j)_{j \geq 0}$ be an upward random walk starting from $0$ with steps drawn from $D_+ = \frac{1+\Delta}{2}\delta_1 + \frac{1-\Delta}{2}\delta_{-1}$. Let $(W_j)_{j \geq 0}$ be a downward random walk starting from $0$ with steps drawn from $D_- = \frac{1-\Delta}{2}\delta_1 + \frac{1+\Delta}{2}\delta_{-1}$. Conditioned on $a^\star = 1$ (resp. $a^\star \neq 1$), the trajectory of the $X^1$ variable can be coupled with $(Z_j)_{j \geq 0}$ (resp. $(W_j)_{j \geq 0}$) as long as it has not reached below $\theta = -\frac{1}{\Delta}$.

Let us first prove $\mathbb{P}(\mathcal{E} | a^\star = 1) = \Omega(1)$. Let $\mathcal{E}_+$ denote the event that $\min_{0 \leq u \leq s} Z_u > -\frac{1}{\Delta}$ and $Z_s \leq s\Delta + \frac{2}{\Delta}$, then $\mathbb{P}(\mathcal{E} | a^\star = 1) = \mathbb{P}(\mathcal{E}_+)$. By Lemma 2.1(b), $\mathbb{P}(\min_{0 \leq u \leq s} Z_u > -\frac{1}{\Delta}) \geq 1 - e^{-2}$. By Hoeffding's inequality, $\mathbb{P}\left(Z_s \geq s\Delta + \frac{2}{\Delta}\right) \leq \exp(-\frac{(2/\Delta)^2}{2s}) \leq e^{-2}$. By union bound, $\mathbb{P}(\mathcal{E}_+) \geq 1 - 2e^{-2} = \Omega(1)$.

Let us now transfer the above bound to $a^\star \neq 1$. Let $\mathcal{E}_-$ denote the event that $\min_{0 \leq u \leq s} W_u > -\frac{1}{\Delta}$ and $W_s \leq s\Delta + \frac{10}{\Delta}$. Then $\mathbb{P}(\mathcal{E} | a^\star \neq 1) = \mathbb{P}(\mathcal{E}_-)$. Let $(z_0, \ldots, z_s)$ be any trajectory of $(Z_j)_{j \geq 0}$ satisfying $\mathcal{E}_+$. Then

$$\mathbb{P}\left((Z_0, \ldots, Z_s) = (z_0, \ldots, z_s)\right) = \left(\frac{1+\Delta}{2}\right)^{(s+z_s)/2} \left(\frac{1-\Delta}{2}\right)^{(s-z_s)/2},$$

and

$$\mathbb{P}\left((W_0,\ldots,W_s)=(z_0,\ldots,z_s)\right) = \left(\frac{1+\Delta}{2}\right)^{(s-z_s)/2}\left(\frac{1-\Delta}{2}\right)^{(s+z_s)/2}$$

$$= \mathbb{P}\left((Z_0,\ldots,Z_s)=(z_0,\ldots,z_s)\right)\left(\frac{1-\Delta}{1+\Delta}\right)^{z_s}.$$

Since for $\Delta = 1 - \Omega(1)$,

$$\left(\frac{1-\Delta}{1+\Delta}\right)^{z_s} = \exp(-O(\Delta z_s)) = \exp(-O(s\Delta^2)) = \Omega(1),$$

a change of measure gives that

$$\mathbb{P}(\mathcal{E}_-) \geq \mathbb{P}(\mathcal{E}_+)\cdot\Omega(1) = \Omega(1).$$

Consequently, $\mathbb{P}(\mathcal{E}) \geq \min\{\mathbb{P}(\mathcal{E}_+),\mathbb{P}(\mathcal{E}_-)\} = \Omega(1)$.

**Proof of** (10). In the sequel we condition on $\mathcal{E}$. Then the next pull the algorithm makes will be to arm 1, i.e., $a_{s+1} = 1$. For any $i \neq 1$ we have $\frac{\mathbb{P}(a^\star=1|\mathcal{H}_s,a_{s+1},\mathcal{E})}{\mathbb{P}(a^\star=i|\mathcal{H}_s,a_{s+1},\mathcal{E})} = (\frac{1+\Delta}{1-\Delta})^{X_s^1} = \Theta(1)$, which implies that $\mathbb{P}(a^\star = 1|\mathcal{H}_s, a_{s+1}, \mathcal{E}) = \Theta\left(\frac{1}{n}\right)$. Therefore,

$$\mathbb{P}(r_{s+1}^{a_{s+1}} = 1|\mathcal{H}_s, a_{s+1}, \mathcal{E}, a^\star = 1) = \frac{1+\Delta}{2},$$

$$\mathbb{P}(r_{s+1}^{a_{s+1}} = 1|\mathcal{H}_s, a_{s+1}, \mathcal{E}) = \frac{1+\Delta}{2}\mathbb{P}(a^\star = 1|\mathcal{H}_s, a_{s+1}, \mathcal{E}) + \frac{1-\Delta}{2}\mathbb{P}(a^\star \neq 1|\mathcal{H}_s, a_{s+1}, \mathcal{E})$$

$$= \frac{1-\Delta}{2} + \Theta\left(\frac{\Delta}{n}\right).$$

It is then clear that

$$\mathrm{KL}\left(P_{r_{s+1}^{a_{s+1}}|\mathcal{H}_s,a_{s+1},\mathcal{E},a^\star=1}\|P_{r_{s+1}^{a_{s+1}}|\mathcal{H}_s,a_{s+1},\mathcal{E}}\right) = \Omega(\Delta^2).$$

Finally,

$$I(a^\star;r_{s+1}^{a_{s+1}}|\mathcal{H}_s,a_{s+1},\mathcal{E}) \geq \mathbb{P}(a^\star = 1)\mathrm{KL}\left(P_{r_{s+1}^{a_{s+1}}|\mathcal{H}_s,a_{s+1},\mathcal{E},a^\star=1}\|P_{r_{s+1}^{a_{s+1}}|\mathcal{H}_s,a_{s+1},\mathcal{E}}\right)$$

$$= \frac{\Omega(\Delta^2)}{n} = \Omega\left(\frac{\Delta^2}{n}\right).$$

## A.2 Proof of (b)

If $t \leq \frac{C}{\Delta^2}$ for some absolute constant $C < \infty$ to be chosen later, then this follows from (a) by $t \geq t_0 = \lfloor\frac{1}{\Delta^2}\rfloor$ and the monotonicity of mutual information. In the following we assume that $t > \frac{C}{\Delta^2}$, and define a variable $X$ measurable with respect to $a^\star$ and a variable $Y$ measurable with respect to $\mathcal{H}_t$. By the data processing inequality, we have $I(a^\star;\mathcal{H}_t) \geq I(X;Y)$. So it suffices to prove the desired lower bounds for $I(X;Y)$.

Let $X = \mathbb{1}\{a^\star \in [m]\}$, with $m := \lceil 0.1t\Delta^2\rceil$. Let $Y$ be the indicator for the following event: the algorithm pulls at most $m$ arms, and the random walk satisfies $X^{\widehat{a}} \geq 0.1t\Delta$ for the action $\widehat{a}$ returned by the algorithm. We will prove the following estimates:

$$\mathbb{P}(Y = 1|X = 1) = \Omega(1), \tag{11}$$

$$\log\frac{\mathbb{P}(Y = 1|X = 1)}{\mathbb{P}(Y = 1|X = 0)} = \Omega(t\Delta^2). \tag{12}$$

**Proof of** (11). The proof is a modification of the proof of Theorem 2.1. For $k \in [m-1]$, let $(W_j^k)_{j\geq 0}$ be $m-1$ independent downward random walks. Let $(Z_j)_{j\geq 0}$ be an upward random walk. Conditioned on $X = 1$, we can couple the $m-1$ bad arm iterations with $W^1,\ldots,W^{m-1}$, and the best arm iteration with $Z$.

Let $T_k$ be the first time that $W^k$ reaches $\theta = -1/\Delta$, and $\mathcal{E}_1$ be the event that $\sum_{k\in[m-1]} T_k \leq 0.2t$. By Lemma 2.1(a) and the same Markov's inequality in the proof of Theorem 2.1, we have $\mathbb{P}(\mathcal{E}_1) \geq 0.5$.

Let $\mathcal{E}_2$ be the event that $\min_{0\leq j\leq t} Z_j > -\frac{1}{\Delta}$. By Lemma 2.1(b), $\mathbb{P}(\mathcal{E}_2) \geq 1 - \exp(-2)$. Let $\mathcal{E}_3$ be the event that $Z_{0.8t} \geq 0.3t\Delta$. By Hoeffding's inequality, $\mathbb{P}(\mathcal{E}_3) \geq 1 - \exp\left(-(0.5t\Delta)^2/(2t)\right) \geq 1 - \exp\left(-C/8\right)$, which is $\geq 0.9$ if we take $C = O(1)$ large enough. Let $\mathcal{E}_4$ be the event that $\min_{0.8t\leq j\leq t}(Z_j - Z_{0.8t}) \geq -0.2t\Delta$. By Lemma 2.1(b), $\mathbb{P}(\mathcal{E}_4) \geq 1 - \exp(-2 \cdot 0.2t\Delta^2) \geq 1 - \exp(-0.4C)$, which is $\geq 0.9$ if we take $C = O(1)$ large enough. Because $\mathcal{E}_2, \mathcal{E}_3, \mathcal{E}_4$ are all monotone events in the steps of $(Z_j)_{j\geq 0}$, we have $\mathbb{P}(\mathcal{E}_2 \cap \mathcal{E}_3 \cap \mathcal{E}_4) \geq \mathbb{P}(\mathcal{E}_2)\mathbb{P}(\mathcal{E}_3)\mathbb{P}(\mathcal{E}_4) = \Omega(1)$. Note that $\mathcal{E}_3 \cap \mathcal{E}_4$ implies that $\min_{0.8t\leq j\leq t} Z_j \geq 0.1t\Delta$.

Via the coupling between the algorithm and the random walks $(W^k_j)_{j\geq 0}$ and $(Z_j)_{j\geq 0}$, when $\mathcal{E}_1, \ldots, \mathcal{E}_4$ all happen, we have $Y = 1$. Therefore $\mathbb{P}(Y = 1|X = 1) \geq \mathbb{P}(\mathcal{E}_1)\mathbb{P}(\mathcal{E}_2\cap\mathcal{E}_3\cap\mathcal{E}_4) = \Omega(1)$.

**Proof of** (12). Fix a history $\mathcal{H}_t$ under which $Y = 1$. Let $s$ denote the number of pulls to arm $\widehat{a}$, then

$$\frac{\mathbb{P}(\mathcal{H}_t|a^\star = \widehat{a})}{\mathbb{P}(\mathcal{H}_t|X = 0)} = \frac{\left(\frac{1+\Delta}{2}\right)^{(s+X^{\widehat{a}})/2}\left(\frac{1-\Delta}{2}\right)^{(s-X^{\widehat{a}})/2}}{\left(\frac{1-\Delta}{2}\right)^{(s+X^{\widehat{a}})/2}\left(\frac{1+\Delta}{2}\right)^{(s-X^{\widehat{a}})/2}} = \left(\frac{1+\Delta}{1-\Delta}\right)^{X^{\widehat{a}}} \geq \exp\left(2\Delta X^{\widehat{a}}\right) \geq \exp(0.2t\Delta^2).$$

Therefore,

$$\frac{\mathbb{P}(\mathcal{H}_t|X = 1)}{\mathbb{P}(\mathcal{H}_t|X = 0)} \geq \frac{\frac{1}{m}\mathbb{P}(\mathcal{H}_t|a^\star = \widehat{a})}{\mathbb{P}(\mathcal{H}_t|X = 0)} \geq \frac{\exp(0.2t\Delta^2)}{\lceil 0.1t\Delta^2\rceil} \geq \exp(0.1t\Delta^2).$$

**Finishing the proof.** Let us prove $\mathrm{KL}(P_{Y|X=1}\|P_Y) = \Omega(t\Delta^2)$ using (11) and (12).

First, we have

$$\begin{aligned}
\frac{\mathbb{P}(Y = 1)}{\mathbb{P}(Y = 1|X = 1)} &= \frac{m}{n} + \left(1 - \frac{m}{n}\right)\frac{\mathbb{P}(Y = 1|X = 0)}{\mathbb{P}(Y = 1|X = 1)} \\
&= \frac{m}{n} + \left(1 - \frac{m}{n}\right)\exp(-\Omega(t\Delta^2)) \\
&\leq 2\max\left\{\frac{m}{n}, \exp(-\Omega(t\Delta^2))\right\}.
\end{aligned}$$

In addition, it holds trivially that

$$\mathbb{P}(Y = 0|X = 1)\log\frac{\mathbb{P}(Y = 0|X = 1)}{\mathbb{P}(Y = 0)} \geq \min_{0\leq x\leq 1} x\log(x) \geq -1.$$

Combining the above displays, we have

$$\begin{aligned}
\mathrm{KL}(P_{Y|X=1}\|P_Y) &= \mathbb{P}(Y = 1|X = 1)\log\frac{\mathbb{P}(Y = 1|X = 1)}{\mathbb{P}(Y = 1)} + \mathbb{P}(Y = 0|X = 1)\log\frac{\mathbb{P}(Y = 0|X = 1)}{\mathbb{P}(Y = 0)} \\
&\geq \Omega(1) \cdot \left(\min\left\{\log\frac{n}{m}, \Omega(t\Delta^2)\right\} - \log 2\right) - 1 \\
&= \Omega(t\Delta^2),
\end{aligned}$$

where the last step follows from $t\Delta^2 \geq C$ for a large enough constant $C$, and that $\log\frac{n}{m} = \Omega(\log n) = \Omega(t\Delta^2)$ by our assumption of $t\Delta^2 \leq \log n$. Finally,

$$I(X; Y) \geq \mathbb{P}(X = 1)\mathrm{KL}(P_{Y|X=1}\|P_Y) = \frac{m}{n} \cdot \Omega(t\Delta^2) = \Omega(t^2\Delta^4).$$

### A.3   Proof of (c)

In fact, the only place where we used the assumption $t\Delta^2 \leq \log n$ in the proof of (b) is to ensure that $\log\frac{n}{m} = \Omega(t\Delta^2)$ for $m = \lceil 0.1t\Delta^2\rceil$. Consequently, for $\log n \leq t\Delta^2 \leq \sqrt{n}$, the argument in the proof of (b) now gives

$$\mathrm{KL}(P_{Y|X=1}\|P_Y) = \Omega(\log n),$$

so that

$$I(a^\star; \mathcal{H}_t) \geq I(X; Y) \geq \mathbb{P}(X = 1)\mathrm{KL}(P_{Y|X=1}\|P_Y) = \frac{m}{n} \cdot \Omega(\log n) = \Omega\left(\frac{t\Delta^2 \log n}{n}\right),$$

which is the desired result of (c). When $\sqrt{n} \leq t\Delta^2 \leq n$, we recall the lower bound of the success probability $p_t$ and Fano's inequality in (6) to get

$$I(a^\star; \mathcal{H}_t) \geq p_t \log \frac{np_t}{e} = \Omega\left(\frac{t\Delta^2}{n} \log \Omega\left(\frac{t\Delta^2}{e}\right)\right) = \Omega\left(\frac{t\Delta^2 \log n}{n}\right),$$

which is again the desired result of (c).

# B  Non-Interactive Case

In this section we prove (4) for non-interactive algorithms, restated as follows for $t \leq \frac{n \log n}{\Delta^2}$:

$$p_{t,\mathrm{NI}}^\star \asymp \frac{t\Delta^2}{n \log(1 + t\Delta^2)}, \qquad I_{t,\mathrm{NI}}^\star \asymp \frac{t\Delta^2}{n}.$$

**Achievability for success probability.**  A non-interactive algorithm is as follows. Pick $m \in [n]$ as the largest integer solution to

$$\frac{4m \log m}{\Delta^2} \leq t, \quad \text{so that } m = \Omega\left(\frac{t\Delta^2}{\log(1 + t\Delta^2)}\right).$$

The learner randomly permutes the action set $[n]$, pulls each of the first $m$ arms $\frac{4 \log m}{\Delta^2}$ times, and outputs the arm with the largest average reward. By the definition of $m$, this algorithm runs in at most $t$ rounds. With probability $\frac{m}{n}$, the best arm $a^\star$ is one of the first $m$ arms. Conditioned on that, the algorithm correctly outputs the best arm $a^\star$ with probability at least $1 - (m-1)p$ by the union bound, where $p$ is the probability that a $\mathrm{Bin}\left(\frac{4 \log m}{\Delta^2}, \frac{1+\Delta}{2}\right)$ random variable is less than or equal to an independent $\mathrm{Bin}\left(\frac{4 \log m}{\Delta^2}, \frac{1-\Delta}{2}\right)$ random variable. By Hoeffding's inequality, we have

$$p \leq \exp\left(-\frac{\Delta^2}{2} \cdot \frac{4 \log m}{\Delta^2}\right) = \frac{1}{m^2}$$

Therefore, the overall success probability of this algorithm is at least

$$\frac{m}{n}\left(1 - \frac{m-1}{m^2}\right) \geq \frac{3m}{4n} = \Omega\left(\frac{t\Delta^2}{n \log(1 + t\Delta^2)}\right).$$

**Achievability for mutual information.**  The above algorithm also attains the optimal mutual information. For $t\Delta^2 \geq C$ with a large enough constant $C = O(1)$, note that Fano's inequality (6) gives

$$I(a^\star; \mathcal{H}_t) \geq p_{t,\mathrm{NI}} \log \frac{np_{t,\mathrm{NI}}}{e} = \Omega\left(\frac{t\Delta^2}{n \log(1 + t\Delta^2)} \log \Omega\left(\frac{t\Delta^2}{\log(1 + t\Delta^2)}\right)\right) = \Omega\left(\frac{t\Delta^2}{n}\right).$$

For $t\Delta^2 \leq 1$, note that $m = 1$ holds in the above algorithm, so it simply pulls a uniformly random arm (say arm 1) for $t$ times. Then the scenario is precisely the same as Appendix A.1, where $a_1 = \cdots = a_t = 1$ always holds, and one can regard the auxiliary random walk $X^1$ as being recorded during the execution of the algorithm. Therefore, Appendix A.1 implies that $I(a^\star; r_{s+1}^{a_{s+1}} | \mathcal{H}_s, a_{s+1}) = \Omega(\frac{\Delta^2}{n})$ for any $s \leq \frac{1}{\Delta^2}$, with the hidden constant independent of $s$. By the chain rule, $I(a^\star; \mathcal{H}_t) = \Omega(\frac{t\Delta^2}{n})$ for all $t \leq \frac{1}{\Delta^2}$.

Finally, for the case $\frac{1}{\Delta^2} \leq t \leq \frac{C}{\Delta^2}$, we simply use the monotonicity of mutual information to obtain

$$I(a^\star; \mathcal{H}_t) \geq I(a^\star; \mathcal{H}_{1/\Delta^2}) = \Omega\left(\frac{1}{n}\right) = \Omega\left(\frac{t\Delta^2}{n}\right),$$

as claimed.

**Converse for mutual information.** Suppose a non-interactive algorithm takes actions $a_1, \ldots, a_t$. Let $r_1^{a_1}, \ldots, r_t^{a_t}$ be the corresponding rewards. Because $I(a^\star; \mathcal{H}_t) = \mathbb{E}_{a_1, \ldots, a_t} I(a^\star; \mathcal{H}_t | a_1, \ldots, a_t)$, we can without loss of generality assume that $a_1, \ldots, a_t$ are deterministic. Then $(a_1, r_1^{a_1}), \ldots, (a_t, r_t^{a_t})$ are independent conditioned on $a^\star$. So

$$I(a^\star; \mathcal{H}_t) \leq \sum_{i=1}^{t} I(a^\star; r_i^{a_i}) = t I(a^\star; r_1^{a_1}).$$

By [Theorem 3.3(a)], we have $I(a^\star; r_1^{a_1}) = O\left(\frac{\Delta^2}{n}\right)$. We thus conclude that $I_{t,\mathrm{NI}}^\star \lesssim \frac{t\Delta^2}{n}$.

**Converse for success probability.** By the converse for mutual information and Fano's inequality [(6)], we have

$$(np_{t,\mathrm{NI}}^\star) \log \frac{np_{t,\mathrm{NI}}^\star}{e} \leq n I_{t,\mathrm{NI}}^\star = O(t\Delta^2).$$

Solving this inequality gives $p_{t,\mathrm{NI}}^\star \lesssim \frac{t\Delta^2}{n \log(1 + t\Delta^2)}$.

## C Deferred Proofs in [Section 4] and More Discussions

### C.1 Proof of [Proposition 4.1]

Consider the sampling strategy where each action is pulled for $t/n$ times (for $t < n$ we simply pull each of the first $t$ arms once). If $t < n$, then for any $\mathbb{P}_i$ and $\mathbb{P}_j$, by data-processing inequality,

$$\mathrm{TV}(\mathbb{P}_i, \mathbb{P}_j) \geq \mathrm{TV}\left(\mathrm{Ber}\left(\frac{1-\Delta}{2}\right), \mathrm{Ber}\left(\frac{1+\Delta}{2}\right)\right) \cdot \mathbb{1}(i \text{ or } j \text{ is pulled})$$

$$= \Delta \cdot \mathbb{1}(i \text{ or } j \text{ is pulled}).$$

For any $i$ and $j$ that are pulled, we have by the triangle inequality that for any reference distribution $\mathbb{P}_0$,

$$\mathrm{TV}(\mathbb{P}_0, \mathbb{P}_i) + \mathrm{TV}(\mathbb{P}_0, \mathbb{P}_j) \geq \mathrm{TV}(\mathbb{P}_i, \mathbb{P}_j) \gtrsim \Delta.$$

This shows that

$$\frac{1}{n} + \inf_{\mathbb{P}_0} \frac{1}{n} \sum_{i=1}^{n} \mathrm{TV}(\mathbb{P}_0, \mathbb{P}_i) \geq \frac{1}{n} + \Omega\left(\frac{t\Delta}{n}\right).$$

If $t \geq n$, without loss of generality we assume that $t/n$ is an integer. For any $\mathbb{P}_i$ and $\mathbb{P}_j$, by data-processing inequality,

$$\mathrm{TV}(\mathbb{P}_i, \mathbb{P}_j) \geq \mathrm{TV}(\mathrm{Bin}(\frac{t}{n}, \frac{1-\Delta}{2}), \mathrm{Bin}(\frac{t}{n}, \frac{1+\Delta}{2})) \gtrsim \sqrt{\frac{t\Delta^2}{n}}.$$

Here the last inequality uses the TV lower bound for two binomial distributions in [Lemma C.2], whose proof is deferred to the end of this section. By the triangle inequality, we have that for any reference distribution $\mathbb{P}_0$,

$$\mathrm{TV}(\mathbb{P}_0, \mathbb{P}_i) + \mathrm{TV}(\mathbb{P}_0, \mathbb{P}_j) \geq \mathrm{TV}(\mathbb{P}_i, \mathbb{P}_j) \gtrsim \sqrt{\frac{t\Delta^2}{n}}.$$

Finally, this shows that for any choice of reference model

$$\frac{1}{n} + \inf_{\mathbb{P}_0} \frac{1}{n} \sum_{i=1}^{n} \mathrm{TV}(\mathbb{P}_0, \mathbb{P}_i) \geq \frac{1}{n} + \Omega\left(\sqrt{\frac{t\Delta^2}{n}}\right).$$

Combining the above two cases completes the proof of [Proposition 4.1].

To complete this section, we include some useful results, and the proof of [(7)] for completeness.

**Lemma C.1.** *For any $k \geq 1$ and $\frac{k - \sqrt{k}}{2} \leq \ell \leq \frac{k + \sqrt{k}}{2}$, we have $\binom{k}{\ell} \geq \frac{2^k}{4\sqrt{k}}$.*

**Proof.** We have

$$\frac{\binom{k}{(k+\sqrt{k})/2}}{\binom{k}{k/2}} = \prod_{i=1}^{\sqrt{k}/2} \frac{k/2 - i}{k/2 + \sqrt{k}/2 - i}$$

$$= \prod_{i=1}^{\sqrt{k}/2} \left(1 - \frac{\sqrt{k}/2}{k/2 + \sqrt{k}/2 - i}\right)$$

$$\geq 1 - \sum_{i=1}^{\sqrt{k}/2} \frac{\sqrt{k}/2}{k/2 + \sqrt{k}/2 - i} \geq \frac{1}{2},$$

where we have used the simple inequality $\prod_{i=1}^{n}(1 - a_i) \geq 1 - \sum_{i=1}^{n} a_i$ for $a_1, \ldots, a_n \in (0,1)$. Thus, we obtain

$$\binom{k}{(k+\sqrt{k})/2} \geq \frac{1}{2}\binom{k}{k/2} \geq \frac{2^k}{4\sqrt{k}},$$

where the last step follows from Stirling's approximation. $\qquad\square$

**Lemma C.2.** *For any $0 < \Delta = 1 - \Omega(1)$, for any integer $k \geq 1$ such that $k\Delta^2 \leq 1$, we have* $\mathrm{TV}\big(\mathrm{Bin}(k, \frac{1-\Delta}{2}), \mathrm{Bin}(k, \frac{1+\Delta}{2})\big) \gtrsim \sqrt{k\Delta^2}$, *where $\mathrm{Bin}(k, p)$ denotes the binomial distribution with parameter $k$ and $p$.*

**Proof.** By Kelbert (2023, Proposition 6), we have

$$\mathrm{TV}\left(\mathrm{Bin}(k, \frac{1-\Delta}{2}), \mathrm{Bin}(k, \frac{1+\Delta}{2})\right) = k\int_{(1-\Delta)/2}^{(1+\Delta)/2} \mathbb{P}(S_{k-1}(u) = \ell - 1)\mathrm{d}u,$$

where $S_{k-1}(u) \sim \mathrm{Bin}(k-1, u)$ and $\ell$ is in the interval $[k(1-\Delta)/2, k(1+\Delta)/2]$. By Lemma C.1, we have for any $u \in [(1-\Delta)/2, (1+\Delta)/2]$ and $\ell \in [k(1-\Delta)/2, k(1+\Delta)/2]$,

$$\mathbb{P}(S_{k-1}(u) = \ell - 1) \geq \frac{1}{4\sqrt{k}}(1-\Delta)^{k-1+k\Delta/2}(1+\Delta)^{k-1-k\Delta/2} \gtrsim \frac{1}{\sqrt{k}}.$$

In turn, we have shown

$$\mathrm{TV}\left(\mathrm{Bin}(k, \frac{1-\Delta}{2}), \mathrm{Bin}(k, \frac{1+\Delta}{2})\right) = k\int_{(1-\Delta)/2}^{(1+\Delta)/2} \mathbb{P}(S_{k-1}(u) = \ell - 1)\mathrm{d}u$$

$$\gtrsim k \cdot \Delta \cdot \frac{1}{\sqrt{k}} = \sqrt{k\Delta^2}.$$

This concludes our proof. $\qquad\square$

**Proof of** (7). This is essentially (Gao et al., 2019, Lemma 3) applied to the star graph with center $\mathbb{P}_0$. For any fixed distribution $\mathbb{P}_0$, we have

$$\mathbb{P}(a_{t+1} = a^\star) = \frac{1}{n}\sum_{i=1}^{n} \mathbb{P}_i(a_{t+1} = a_i) \leq \frac{1}{n}\sum_{i=1}^{n}(\mathbb{P}_0(a_{t+1} = a_i) + \mathrm{TV}(\mathbb{P}_0, \mathbb{P}_i))$$

$$= \frac{1}{n} + \frac{1}{n}\sum_{i=1}^{n} \mathrm{TV}(\mathbb{P}_0, \mathbb{P}_i).$$

Then, by taking infimum over the distribution $\mathbb{P}_0$, we obtain the desired result.

### C.2   Proof of Proposition 4.2

Consider the following learner under the two-phase model: The learner computes the total reward for each arm in the first phase, and then queries the $m = \lceil t\Delta^2 \rceil$ arms with the highest total reward in the interactive phase. W.l.o.g., assume $a^\star = n$ and that the total rewards for each arm are $R_1, \ldots, R_n$. Clearly the learner succeeds if $R_n$ is among the largest $m$ numbers in $R_1, \ldots, R_n$.

Define the threshold $u^\star > 0$ as the solution to

$$\frac{2}{1+u^\star} \exp\left(-\frac{1}{\Delta^2}\mathrm{KL}\left(\mathrm{Ber}(\frac{1-u^\star\Delta}{2})\,\Big\|\,\mathrm{Ber}(\frac{1+\Delta}{2})\right)\right) = \frac{m}{2n}. \tag{13}$$

By Pinsker's inequality, we see from (13) that $u^\star = O(\sqrt{\log \frac{n}{m}})$. As $\Delta = o(\frac{1}{\sqrt{\log n}})$, we have $u^\star\Delta = o(1)$ and therefore

$$\mathrm{KL}\left(\mathrm{Ber}(\frac{1-u^\star\Delta}{2})\,\Big\|\,\mathrm{Ber}(\frac{1+\Delta}{2})\right) \asymp (u^\star+1)^2\Delta^2,$$

from which we readily conclude from (13) that $u^\star = \Theta(\sqrt{\log \frac{n}{m}})$. In addition, since $m = o(n)$ and $\Delta = o(\frac{1}{\sqrt{\log n}})$, for large enough $n$ we have $2 \le u^\star \le \frac{1}{2\Delta}$.

Next we invoke accurate tail estimates for the binomial distribution in Lemma C.3. Since $R_i \sim \mathrm{Bin}(\frac{1}{\Delta^2}, \frac{1-\Delta}{2})$ for any $i \in [n-1]$, the upper bound in Lemma C.3 gives

$$\mathbb{P}(R_i \ge \frac{1+u^\star\Delta}{2\Delta^2}) \le \frac{m}{2n}, \qquad i \in [n-1].$$

Since $R_1, \cdots, R_{n-1}$ are independent, the Chernoff bound gives

$$\mathbb{P}\Big(\underbrace{\text{There are more than } m-1 \text{ suboptimal arms with total rewards larger than } \frac{1+u^\star\Delta}{2\Delta^2}}_{=:\mathcal{E}}\Big)$$

$$= \mathbb{P}\left(\frac{1}{n-1}\sum_{i=1}^{n-1}\mathbb{1}\left(R_i \ge \frac{1+u^\star\Delta}{2\Delta^2}\right) \ge \frac{m}{n-1}\right) \le \left(\frac{e}{4}\right)^{m/2} = 1 - \Omega(1).$$

This implies that $\mathbb{P}(\mathcal{E}^c) = \Omega(1)$. Moreover, by the lower bound in Lemma C.3, we have

$$\mathbb{P}\left(R_n \ge \frac{1+u^\star\Delta}{2\Delta^2}\right) \gtrsim \frac{1}{1+u^\star}\exp\left(-\frac{1}{\Delta^2}\mathrm{KL}\left(\mathrm{Ber}(\frac{1-u^\star\Delta}{2})\,\Big\|\,\mathrm{Ber}(\frac{1-\Delta}{2})\right)\right).$$

By simple algebra,

$$\mathrm{KL}\left(\mathrm{Ber}(\frac{1-u^\star\Delta}{2})\,\Big\|\,\mathrm{Ber}(\frac{1-\Delta}{2})\right) = \mathrm{KL}\left(\mathrm{Ber}(\frac{1-u^\star\Delta}{2})\,\Big\|\,\mathrm{Ber}(\frac{1+\Delta}{2})\right) - u^\star\Delta\log\frac{1+\Delta}{1-\Delta}$$

$$\le \mathrm{KL}\left(\mathrm{Ber}(\frac{1-u^\star\Delta}{2})\,\Big\|\,\mathrm{Ber}(\frac{1+\Delta}{2})\right) - u^\star\Delta^2.$$

This, in turn, gives us

$$\mathbb{P}\left(R_n \ge \frac{1+u^\star\Delta}{2\Delta^2}\right) \gtrsim \frac{1}{1+u^\star}\exp\left(-\frac{1}{\Delta^2}\mathrm{KL}\left(\mathrm{Ber}(\frac{1-u^\star\Delta}{2})\,\Big\|\,\mathrm{Ber}(\frac{1+\Delta}{2})\right)\right)\cdot e^{u^\star}$$

$$\overset{(13)}{=} \frac{m}{4n}e^{u^\star}.$$

Altogether, by independence of $\mathcal{E}^c$ and $R_n$, we have shown that

$$p_t^\star \gtrsim \mathbb{P}\left(\mathcal{E}^c \text{ and } R_n \ge \frac{1+u^\star\Delta}{2\Delta^2}\right) \gtrsim \frac{t\Delta^2}{n}\exp\left(\Omega\left(\sqrt{\log\frac{n}{t\Delta^2}}\right)\right) = \omega\left(\frac{t\Delta^2}{n}\right),$$

where we recall that $u^\star = \Theta(\sqrt{\log\frac{n}{m}})$. This concludes our proof.

**Lemma C.3.** *Let $\Delta \in (0, \frac{1}{4}]$, and $u \in [2, \frac{1}{2\Delta}]$. For $X \sim \mathrm{Bin}(\frac{1}{\Delta^2}, \frac{1-\Delta}{2})$ and $Y \sim \mathrm{Bin}(\frac{1}{\Delta^2}, \frac{1+\Delta}{2})$, it holds that*

$$\begin{cases} \mathbb{P}(X \ge \frac{1+u\Delta}{2\Delta^2}) \le \frac{2}{1+u}\exp\left(-\frac{1}{\Delta^2}\mathrm{KL}(\mathrm{Ber}(\frac{1-u\Delta}{2})\|\mathrm{Ber}(\frac{1+\Delta}{2}))\right) \\ \mathbb{P}(Y \ge \frac{1+u\Delta}{2\Delta^2}) \gtrsim \frac{1}{1+u}\exp\left(-\frac{1}{\Delta^2}\mathrm{KL}(\mathrm{Ber}(\frac{1-u\Delta}{2})\|\mathrm{Ber}(\frac{1-\Delta}{2}))\right) \end{cases}.$$

**Proof.** By (Zhu et al., 2022, Theorem 2.1 and 2.2), we have

$$\mathbb{P}\left(X \ge \frac{1+u\Delta}{2\Delta^2}\right) \le \frac{4\Delta L\left(\frac{1}{\Delta^2}, \frac{1-u\Delta}{2\Delta^2}, \frac{1+\Delta}{2}\right)}{\sqrt{2\pi(1-u^2\Delta^2)}}\exp\left(-\frac{1}{\Delta^2}\mathrm{KL}\left(\mathrm{Ber}(\frac{1-u\Delta}{2})\,\Big\|\,\mathrm{Ber}(\frac{1+\Delta}{2})\right)\right),$$

where the function $L(k, x, p)$ is defined as

$$L(k, x, p) := \frac{x + 1 - kp + \sqrt{(kp - x + 1)^2 + 4(1 - p)x}}{2}$$

$$= 1 + \frac{2(1 - p)x}{kp + 1 - x + \sqrt{(kp + 1 - x)^2 + 4(1 - p)x}}.$$

We thus estimate

$$L\left(\frac{1}{\Delta^2}, \frac{1 - u\Delta}{2\Delta^2}, \frac{1 + \Delta}{2}\right) = 1 + \frac{2\frac{1 - \Delta}{2}\frac{1 - u\Delta}{2\Delta^2}}{1 + \frac{1 + u}{2\Delta} + \sqrt{\left(1 + \frac{1 + u}{2\Delta}\right)^2 + \frac{(1 - \Delta)(1 - u\Delta)}{\Delta^2}}}$$

$$\leq 1 + \frac{(1 - \Delta)(1 - u\Delta)}{2\Delta^2\left(\frac{1 + u}{\Delta}\right)}$$

$$= \frac{(1 + \Delta)(1 + u\Delta)}{2\Delta(1 + u)} < \frac{1}{\Delta(1 + u)},$$

where the last step uses $(1 + \Delta)(1 + u\Delta) \leq \frac{5}{4} \cdot \frac{3}{2} < 2$. Thus combining with the assumption that $u \leq \frac{1}{2\Delta}$, we have

$$\mathbb{P}\left(X \geq \frac{1 + u\Delta}{2\Delta^2}\right) \leq \frac{4\Delta L\left(\frac{1}{\Delta^2}, \frac{1 - u\Delta}{2\Delta^2}, \frac{1 + \Delta}{2}\right)}{\sqrt{2\pi(1 - u^2\Delta^2)}} \exp\left(-\frac{1}{\Delta^2}\mathrm{KL}\left(\mathrm{Ber}(\frac{1 - u\Delta}{2})\,\middle\|\,\mathrm{Ber}(\frac{1 + \Delta}{2})\right)\right)$$

$$\leq \frac{2}{1 + u}\exp\left(-\frac{1}{\Delta^2}\mathrm{KL}\left(\mathrm{Ber}(\frac{1 - u\Delta}{2})\,\middle\|\,\mathrm{Ber}(\frac{1 + \Delta}{2})\right)\right)$$

as desired. Similarly, we have for the other side by (Zhu et al., 2022, Theorem 2.2),

$$\mathbb{P}\left(Y \geq \frac{1 + u\Delta}{2\Delta^2}\right) \geq \frac{2\Delta L\left(\frac{1}{\Delta^2}, \frac{1 - u\Delta}{2\Delta^2}, \frac{1 - \Delta}{2}\right)}{\sqrt{8(1 - u^2\Delta^2)}} \exp\left(-\frac{1}{\Delta^2}\mathrm{KL}\left(\mathrm{Ber}(\frac{1 - u\Delta}{2})\,\middle\|\,\mathrm{Ber}(\frac{1 - \Delta}{2})\right)\right).$$

We lower bound the function $L$ as

$$L\left(\frac{1}{\Delta^2}, \frac{1 - u\Delta}{2\Delta^2}, \frac{1 - \Delta}{2}\right) = 1 + \frac{\frac{(1 + \Delta)(1 - u\Delta)}{2\Delta^2}}{1 + \frac{u - 1}{2\Delta} + \sqrt{\left(1 + \frac{u - 1}{2\Delta}\right)^2 + \frac{(1 + \Delta)(1 - u\Delta)}{\Delta^2}}}$$

$$\gtrsim \frac{(1 + \Delta)(1 - u\Delta)}{\Delta^2\left(\frac{1 + u}{\Delta}\right)} \gtrsim \frac{1}{\Delta(1 + u)}.$$

Thus, we have

$$\mathbb{P}\left(Y \geq \frac{1 + u\Delta}{2\Delta^2}\right) \gtrsim \frac{2\Delta L\left(\frac{1}{\Delta^2}, \frac{1 - u\Delta}{2\Delta^2}, \frac{1 - \Delta}{2}\right)}{\sqrt{8(1 - u^2\Delta^2)}} \exp\left(-\frac{1}{\Delta^2}\mathrm{KL}\left(\mathrm{Ber}(\frac{1 - u\Delta}{2})\,\middle\|\,\mathrm{Ber}(\frac{1 - \Delta}{2})\right)\right)$$

$$\gtrsim \frac{1}{1 + u}\exp\left(-\frac{1}{\Delta^2}\mathrm{KL}\left(\mathrm{Ber}(\frac{1 - u\Delta}{2})\,\middle\|\,\mathrm{Ber}(\frac{1 - \Delta}{2})\right)\right).$$

This concludes our proof. $\qquad\square$

### C.3 Proof of Proposition 4.3

We present an algorithm (cf. Algorithm 3) that achieves an optimal success probability but suboptimal mutual information when $t \in \frac{\omega(1)}{\Delta^2} \cap \frac{n^{o(1)}}{\Delta^2}$.

**Success probability.** The proof is a modification of the proof of Theorem 2.1. The main difference between Algorithm 3 and Algorithm 1 is that in addition to the lower threshold $\theta_l$, we now have an upper threshold $\theta_r$. When the random walk for an arm reaches $\theta_r$, we immediately return it as our estimate for the best arm instead of continuing pulling it. Another minor difference is that, Algorithm 3 only pulls the first $w := \lceil t\Delta^2 \rceil$ arms, even if the time budget has not been exhausted.

By the same reasoning as the proof of Theorem 2.1, Algorithm 3 can be restated as follows.

---

**Algorithm 3** MODIFIEDSEQUENTIALPROBABILITYRATIOTEST$(A, t, \Delta)$

---
1: **input:** action set $A$, number of rounds $t$, noise parameter $\Delta$
2: **output:** an estimate of the best arm $\widehat{a} \in A$
3: Permute $A$ uniformly at random. Relabel elements of $A$ as $1, \ldots, n$ where $n = |A|$.
4: $w \leftarrow \lceil t\Delta^2 \rceil, \theta_l \leftarrow -1/\Delta, \theta_r \leftarrow \frac{\log w}{\log \frac{1+\Delta}{1-\Delta}}, s \leftarrow 0$
5: **for** $i = 1$ to $w$ **do**             ▷ Only explores the first $w$ arms
6:   $X^i \leftarrow 0$
7:   **while** true **do**
8:    **if** $s = t$ **then return** $\widehat{a} = i$
9:    Pull action $i$ and receive reward $r_t^i \in \{0, 1\}$
10:    $X^i \leftarrow X^i + 2r_t^i - 1, s \leftarrow s + 1$
11:    **if** $X^i \geq \theta_r$ **then return** $\widehat{a} = i$      ▷ Early stops for a promising estimate
12:    **if** $X^i \leq \theta_l$ **then break**
13: **return** $\widehat{a} = 1$

---

1. Permute the arms uniformly at random.

2. For each arm $i$, if arm $i$ is the best arm, let $(X_j^i)_{j \geq 0}$ be the upward random walk (starting from 0 with steps drawn from $D_+$); otherwise let $(X_j^i)_{j \geq 0}$ be the downward random walk (starting from 0 with steps drawn from $D_-$). All these random walks are independent.

3. Let $T_i$ be the first time that $X_{T_i}^i \leq \theta_l$ and $U_i$ be the first time that $X_{U_i}^i \geq \theta_r$. Let $i$ be the smallest index such that $\sum_{k \in [i]} T_k > t$ and $j$ be the smallest index such that $U_j < \infty$. If either $i$ or $j$ exists, return arm $\min\{i, j\}$. Otherwise return arm 1.

Let $m = \lceil 0.1t\Delta^2 \rceil$. Because $t \in \frac{\omega(1)}{\Delta^2} \cap \frac{n^{o(1)}}{\Delta^2}$, we have $m \in \omega(1) \cap n^{o(1)}$. Let $\mathcal{E}_1$ be the event that the best arm is among the first $m$ arms (after random permutation). Let $\mathcal{E}_2$ be the event that $\sum_{i \in [m] \setminus \{i^\star\}} T_i \leq t$, where $i^\star$ is the optimal arm after the random permutation. Let $\mathcal{E}_3$ be the event that $U_i = \infty$ for $i \in [m] \setminus \{i^\star\}$. Let $\mathcal{E}_4$ be the event that $T_{i^\star} = \infty$. If $\mathcal{E}_1 \cap \mathcal{E}_2 \cap \mathcal{E}_3 \cap \mathcal{E}_4$ happens, then the algorithm returns the correct answer after $t$ rounds. In the following, we prove that $\mathbb{P}(\mathcal{E}_1 \cap \mathcal{E}_2 \cap \mathcal{E}_3 \cap \mathcal{E}_4) = \Omega\left(\frac{m}{n}\right)$.

Clearly, $\mathbb{P}(\mathcal{E}_1) = \frac{m}{n}$. By the same proof as Theorem 2.1, we have $\mathbb{P}(\mathcal{E}_2 | \mathcal{E}_1) \geq 0.9$ and $\mathbb{P}(\mathcal{E}_4 | \mathcal{E}_1) \geq 1 - e^{-2}$. It remains to consider $\mathcal{E}_3$. By Lemma 2.1(b), for $i \neq i^\star$, $\mathbb{P}(U_i < \infty) \leq (\frac{1-\Delta}{1+\Delta})^{\theta_r} = \frac{1}{w}$, with $w := \lceil t\Delta^2 \rceil$. By a union bound, $\mathbb{P}(\mathcal{E}_3^c | \mathcal{E}_1) \leq \frac{m-1}{w} \leq 0.1$. Therefore

$$\mathbb{P}(\mathcal{E}_1 \cap \mathcal{E}_2 \cap \mathcal{E}_3 \cap \mathcal{E}_4) \geq \mathbb{P}(\mathcal{E}_1)(1 - \mathbb{P}(\mathcal{E}_2^c | \mathcal{E}_1) - \mathbb{P}(\mathcal{E}_3^c | \mathcal{E}_1) - \mathbb{P}(\mathcal{E}_4^c | \mathcal{E}_1)) = \Omega\left(\frac{m}{n}\right).$$

This completes the proof.

**Mutual information.** The analysis relies on an explicit expression for the posterior distribution $P_{a^\star | \mathcal{H}_t}$ of the optimal arm given observations. Given $\mathcal{H}_t$, let $s_i$ denote the number of pulls to arm $i$ and let $c_i$ denote the value of $X^i$ when the algorithm returns, for each $i \in [n]$. Then we have

$\mathbb{P}(\mathcal{H}_t | a^\star = i)$

$$= \left(\prod_{j \in [n] \setminus \{i\}} \left(\frac{1-\Delta}{2}\right)^{(s_j + c_j)/2} \left(\frac{1+\Delta}{2}\right)^{(s_j - c_j)/2}\right) \left(\left(\frac{1+\Delta}{2}\right)^{(s_i + c_i)/2} \left(\frac{1-\Delta}{2}\right)^{(s_i - c_i)/2}\right)$$

$$\propto \left(\frac{1+\Delta}{1-\Delta}\right)^{c_i} = \exp\left(c_i \log \frac{1+\Delta}{1-\Delta}\right).$$

By Bayes' theorem, the posterior distribution $P_{a^\star | \mathcal{H}_t}$ satisfies

$$P_{a^\star | \mathcal{H}_t}(i) \propto \exp\left(c_i \log \frac{1+\Delta}{1-\Delta}\right). \tag{14}$$

In particular, the posterior distribution depends only on $c_i$'s, but not on $s_i$'s.

Let $i_0$ be the last arm pulled by the algorithm (before it returns). Let $U \subseteq [n]$ be the set of arms that were pulled, excluding $i_0$. Let $V \subseteq [n]$ be the set of arms that were not pulled. Note that $|U| \leq w - 1$ and $[n] = U \cup V \cup \{i_0\}$. Next we compute the posterior distribution $P_{a^\star | \mathcal{H}_t}$ based on (14). For $i \in U$, we have $c_i = \lfloor \theta_l \rfloor$, and

$$\exp\left(c_i \log \frac{1+\Delta}{1-\Delta}\right) = \exp\left(\lfloor \theta_l \rfloor \log \frac{1+\Delta}{1-\Delta}\right) = \exp\left(\Theta(1)\right) = \Theta(1).$$

In the middle step we have used the assumption $\Delta = 1 - \Omega(1)$. For $i \in V$, we simply have $c_i = 0$. For the arm $i_0$, we have $\theta_l < c_{i_0} \leq \lceil \theta_r \rceil$, so

$$\Omega(1) = \exp\left(c_{i_0} \log \frac{1+\Delta}{1-\Delta}\right) \leq \exp\left(\lceil \theta_r \rceil \log \frac{1+\Delta}{1-\Delta}\right) \leq \exp\left(\log w + \log \frac{1+\Delta}{1-\Delta}\right) = O(w).$$

Consequently, according to (14), the posterior distribution of $a^\star$ given $\mathcal{H}_t$ is

$$P_{a^\star | \mathcal{H}_t} = \Big( \underbrace{\frac{a}{Z}, \dots, \frac{a}{Z}}_{\text{arms in } U}, \underbrace{\frac{b}{Z}}_{\text{arm } i_0}, \underbrace{\frac{1}{Z}, \dots, \frac{1}{Z}}_{\text{arms in } V} \Big),$$

where $a = \exp\left(\lfloor \theta_l \rfloor \log \frac{1+\Delta}{1-\Delta}\right) = \Theta(1)$, $b = \exp\left(c_{i_0} \log \frac{1+\Delta}{1-\Delta}\right) \in \Omega(1) \cap O(w)$, and

$$Z = ka + b + (n - k - 1)$$

is the normalizing factor, with $k := |U| \leq w - 1$.

By the above expression of $P_{a^\star | \mathcal{H}_t}$, we have

$$
\begin{aligned}
\mathrm{KL}\left(P_{a^\star | \mathcal{H}_t} \| \mathrm{Unif}([n])\right) &= k\frac{a}{Z}\log\frac{na}{Z} + \frac{b}{Z}\log\frac{nb}{Z} + (n - k - 1)\log\frac{n}{Z} \\
&= \log\frac{n}{Z} + k\frac{a}{Z}\log a + \frac{b}{Z}\log b \\
&\leq \left|\frac{Z}{n} - 1\right| + \frac{ka\log a + b\log b}{Z} \\
&\leq \frac{k|a-1| + |b-1|}{n} + \frac{ka\log a + b\log b}{Z} \\
&\lesssim \frac{w\log w}{n},
\end{aligned}
$$

so that

$$I(a^\star; \mathcal{H}_t) = \mathbb{E}_{\mathcal{H}_t}\left[\mathrm{KL}\left(P_{a^\star | \mathcal{H}_t} \| \mathrm{Unif}([n])\right)\right] = O\left(\frac{w\log w}{n}\right) = O\left(\frac{t\Delta^2 \log(t\Delta^2)}{n}\right),$$

which is the claimed result.

### C.4 Proof of Proposition 4.4

**Achievability.** We run Algorithm 1 with $\frac{n}{\Delta^2}$ rounds with probability $\frac{t\Delta^2}{n}$ and return a uniformly random arm otherwise. The expected number of pulls is $\frac{n}{\Delta^2} \cdot \frac{t\Delta^2}{n} = t$. When the former happens, we get $\Omega(1)$ success probability and $\Omega(\log n)$ mutual information by Theorem 1.1. When the latter happens, we get $\frac{1}{n}$ success probability and $0$ mutual information. So the expected success probability is $\Omega\left(\frac{t\Delta^2}{n} + \frac{1}{n}\left(1 - \frac{t\Delta^2}{n}\right)\right) = \Omega\left(\max\left\{\frac{1}{n}, \frac{t\Delta^2}{n}\right\}\right)$ and the expected mutual information is $\Omega\left(\frac{t\Delta^2 \log n}{n}\right)$.

**Converse for success probability.** By Theorem 1.1, there exists $c > 0$ such that any algorithm that always makes at most $\frac{cn}{\Delta^2}$ pulls has success probability at most $0.1$. Now suppose we have an algorithm $\mathcal{A}$ that makes $\frac{0.1cn}{\Delta^2}$ queries in expectation. By Markov's inequality, the probability that $\mathcal{A}$ makes more than $\frac{cn}{\Delta^2}$ queries is at most $0.1$. Let $\mathcal{A}'$ be the algorithm that runs $\mathcal{A}$, but stops and returns a uniformly random arm when $\mathcal{A}$ is about to make more than $\frac{cn}{\Delta^2}$ queries. Then $\mathcal{A}'$ always makes at most $\frac{cn}{\Delta^2}$ queries, thus has success probability at most $0.1$. By union bound, $\mathcal{A}$ has success probability at most $0.2$. We have proved that $p_{t,\mathrm{E}}^\star \leq 0.2$ for any algorithm that makes at most $\frac{0.1cn}{\Delta^2}$ queries in expectation. The rest of the proof uses the boosting argument in Section 3.2 and is omitted.

### C.5  A conjecture for the stopping time setting

We conjecture that our achievability result for $I_{t,\mathrm{E}}^{\star}$ in [Proposition 4.4](#) is tight (i.e., $I_{t,\mathrm{E}}^{\star} \lesssim \frac{t\Delta^2 \log n}{n}$) and leave this as an open problem.

Using the randomization argument in the above proof, one can show that $\frac{1}{t} I_{t,\mathrm{E}}^{\star}$ is a non-increasing function in $t$. Therefore, our conjecture is equivalent to that $\lim_{t\to 0^+} \frac{1}{t} I_{t,\mathrm{E}}^{\star} \lesssim \frac{\Delta^2 \log n}{n}$.

Let us briefly discuss the difficulty in proving a tight converse for $I_{t,\mathrm{E}}^{\star}$. The most natural idea is to adapt our proof of [Theorem 3.1(a)](#) to the stopping time setting. During the proof, we need to upper bound $\inf_{\overline{P}_{\mathcal{H}_t}} \mathbb{E}_{a^\star}\big[\mathrm{KL}\big(\overline{P}_{\mathcal{H}_t} \| P_{\mathcal{H}_t | a^\star}\big)\big]$ for a model $\overline{P}_{\mathcal{H}_t}$ of our choice. For the fixed time budget case, we choose $\overline{P}_{\mathcal{H}_t}$ to be the dummy model where all arms have reward distribution $\mathrm{Ber}\big(\frac{1-\Delta}{2}\big)$. However, for the stopping time case, it is not guaranteed that the expected number of pulls under this dummy model is still at most $t$. In fact, there exist algorithms that have expected number of pulls $t$ under the actual model, but have infinite expected number of pulls under the dummy model. Therefore, it is unclear how to adapt the proof of [Theorem 3.1(a)](#) to the stopping time case.

