# OpenReview forum: "Evolution of Information in Interactive Decision Making:  A Case Study for Multi-Armed Bandits"
_NeurIPS.cc/2025/Conference — NeurIPS 2025 poster_

### Official Review · Reviewer_TiQo · 2025-07-02

**Clarity:** 3
**Significance:** 2
**Originality:** 3
**Rating:** 4
**Confidence:** 4

**Summary:**

In the context of best arm identification with bandit feedback, the paper establishes bounds on the probability to identify the best arm, and the mutual information between the index of the best arm and the sequence of actions and rewards. The bounds remains tights in the small budget regims ($t \leqslant \frac{1}{\Delta^2}$, $t \leqslant \frac{\log n}{\Delta^2}$, and $t \leqslant \frac{n}{\Delta^2}$).

**Questions:**

Current achievability analysis is based on a specific form of the reward distributions and on the knowledge of the gap $\Delta$. Can we expect an algorithm which would require less assumptions?

**Ethical Concerns:**

["NO or VERY MINOR ethics concerns only"]

**Final Justification:**

The reviews and discussions did clarify the role played by Algorithms 1 and 2, and therefore the position of the paper.
Also, the authors did express their willingness to update the paper in line with these discussions.

Overall, it pushes me to raise the score of the paper. But up to some limit, because, like other reviewers, I would appreciate to feel a more general applicability of the presented results: other bandit instances, improvement of bandit algorithms, ...

**Limitations:**

yes

**Quality:**

3

**Strengths And Weaknesses:**

* S1: The analysis showcases the challenges of the small budget regims.
* S2: The paper includes thorough discussions on the interpretation of these results
* S3: The paper includes thorough comparisons of these results to state of the art in several related fields of research.

* W1: The algorithm used to showcase the achievability of the bound  requires a the rewards to follow Bernoulli distributions of means close to 0.5, with a known gap $\Delta$, which questions the generality of the bound.
* (small/related) W2: In its current form, this algorithm does not offer to choose its error probability.
* (small/related) W3: In its current form, the error probability in the achievability bound takes the form $C.\frac{t\Delta^2}{n}$ where $C$ is a universal constant.

### Remarks
* It's hard to get a proper intuition of $m$ as an "effective number of arms pulleds by the algorithm" at Line 107. The problem arises from the fact that there is no indication on the form of the algorithm before, and the algorithm is somehow surprising.
* typo at L107: it should be $m=\lceil t\Delta^2\rceil$
* L132: is it binary entropy or natural entropy (given that you use $1 = \log e$ to get Equation (6))
* Algorithm 2 is displayed in the core of the paper but neither cited nore discussed. I think the paragraph introducing Section C.1 would be more informative here.

---

> ### Author Rebuttal · Authors · 2025-07-30
>
> We sincerely thank you for your thoughtful feedback and valuable suggestions.
>
> **General response**
>
> We thank the reviewers for suggesting more topics of practical motivation and broader relevance. We will include more discussions accordingly. Indeed, our paper is not really about new algorithms but more a conceptual study about how the information gain (and success probability) behaves in adaptive settings in comparison to non-adaptive ones. The scenario we choose is a natural choice for studying mutual information in adaptive settings with general implications, for the following reasons:
>
> - Upper bounding mutual information is difficult for general adaptive settings and inspires recent works on ridge bandits [1] or linear bandits [2];
>
> - Mutual information is conjectured to be key for understanding sample complexity [2], in addition to recent DEC progresses, which rely on two-point lower bounds; and
>
> - The specific multi-armed bandit case we use is standard for lower bound constructions, reflecting core challenges of the problem. In particular, here the simplicity of this setting is a strength, for it reveals a surprising phenomenon and underscores the need for new analytical tools even in this well-understood baseline scenario.
>
> We make the following contributions by studying this special case:
>
> - We propose simple yet powerful tools to upper bound success probability and mutual information in multi-armed bandits, a canonical problem with a different structure from [1, 2], rendering their techniques inapplicable;
> We reveal a decoupling between mutual information and optimal sample complexity, providing a surprising limitation of the general program in [2].
>
> - We would also like to note that the algorithm is not really our focus: an algorithm which randomly samples $\lceil t\Delta^2 \rceil$ arms and runs a bandit algorithm with optimal regret on them (such as the median elimination algorithm in [3]) would achieve the same success probability. Instead, we use an algorithm that is intuitive (making the source of the logarithmic improvement easy to explain) and simple to analyze the mutual information.
>
> **Specific response**
>
> The algorithm used to showcase the achievability of the bound requires the rewards to follow Bernoulli distributions of means close to 0.5, with a known gap $\Delta$ which questions the generality of the bound.
>
> - Please refer to our general response. Our paper is not really about new algorithms but more a conceptual study about how the information gain (and success probability) behaves in adaptive settings in comparison to non-adaptive ones.
>
> In its current form, this algorithm does not offer to choose its error probability.
>
> - Our converse results show that ANY algorithm (including those that can choose its error probability) cannot achieve a success probability better than our bound. Our achievability results use an algorithm which does not need to choose its error probability.
>
> In its current form, the error probability in the achievability bound takes the form has a universal constant.
>
> - Rate-optimal results (i.e. optimal within universal constants) are standard practice in the bandit literature. For example, most regret guarantees involve universal constants, and best arm identification with a fixed confidence will also involve a universal constant in the sample complexity bound.
>
> Current achievability analysis is based on a specific form of the reward distributions and on the knowledge of the gap Delta. Can we expect an algorithm which would require less assumptions?
>
> - As in our general response, our focus is a case study about how the information gain (and success probability) could have surprising behaviors in adaptive settings, rather than propose an algorithm for this toy case. We choose to use our random walk algorithm that is intuitive (making the source of the logarithmic improvement easy to explain) and simple to analyze the mutual information.
>
> [1] Rajaraman, N., Han, Y., Jiao, J., & Ramchandran, K. (2024). Statistical complexity and optimal algorithms for nonlinear ridge bandits. The Annals of Statistics, 52(6), 2557-2582.
>
> [2] Chen, F., Foster, D. J., Han, Y., Qian, J., Rakhlin, A., & Xu, Y. (2024). Assouad, Fano, and Le Cam with Interaction: A Unifying Lower Bound Framework and Characterization for Bandit Learnability. Advances in Neural Information Processing Systems, 37, 75585-75641.
>
> [3] Even-Dar, E., Mannor, S., & Mansour, Y. (2002, June). PAC bounds for multi-armed bandit and Markov decision processes. In International Conference on Computational Learning Theory (pp. 255-270). Berlin, Heidelberg: Springer Berlin Heidelberg.
>
> [4] Garivier, A., Ménard, P., & Stoltz, G. (2019). Explore first, exploit next: The true shape of regret in bandit problems. Mathematics of Operations Research, 44(2), 377-399.

---

> > ### Comment · Reviewer_TiQo · 2025-08-05
> >
> > Thanks for the rebuttal, it clarifies the role played by Algorithms 1 and 2, and therefore the position of the paper.
> >
> > In terms of bounds, in my opinion the main contribution of the paper (in its current form) is a lower-bound. The corresponding upper-bound should
> > * either be presented as much more limited in terms of general usage as it requires a restrictive form of distribution and the knowledge of $\Delta$ (and hence Theorem 1.1 should be split in 2);
> > * or the fact that *median elimination algorithm* achieves the same regret (including at small regime $t$), should be detailed and/or proven in the paper.
> >
> > Algorithms 1 and 2 remains clever tools for the proof and are indeed part of the contributions of the paper. But hey should be clearly presented as such. Currently, while reading the paper, we have the feeling that they are usable bandit algorithms. I see at least 4 reasons for these misunderstanding and I hope you may work on them:
> > * the habits in bandit literature (on which you cannot act, but you could clarify this unusual role quickly in the paper);
> > * Theorem 1.1 which merges the lowerbounds and their achievability, and which is unclear on the fact that the achievability only stands for know $\Delta$;
> > * the fact that Algorithm 2, and achievabilities are presented  in Figure 1 at the same level as other contributions  (their boxes could for example be greyed, with a clarification in the legend on their more limited applicability)
> > * the fact that this role as a tool is not clarified quickly enough in the paper (I may have misses it).
> >
> > Overall, this clarification pushes me to raise the score of the paper. But up to some limit, because, like other reviewers, I would appreciate to feel a more general applicability of the presented results: other bandit instances, improvement of bandit algorithms, ...

---

> > > ### Author Response · Authors · 2025-08-05
> > >
> > > Thank you so much for your detailed response and summary! We completely agree and will do a better job in clarifying our paper's unconventional role in the revision (such as greying out the achievability part for $p_t^\star$ in Figure 1, as you suggested).

---

### Official Review · Reviewer_VwpF · 2025-07-03

**Clarity:** 3
**Significance:** 2
**Originality:** 3
**Rating:** 4
**Confidence:** 3

**Summary:**

The paper's starting point is an interesting observation on multi-armed bandits.
Assume that we have $n$ arms with Bernoulli rewards,
where the optimal arm has parameter $(1+\Delta)/2$ and all other arms
have parameters $(1-\Delta)/2$. It is known that the sample complexity of
finding the best arm in the multi-armed bandit problem is (ignoring constant factors
here and throughout)
$n/\Delta^2$. However, if we were to consider a non-interactive setting (i.e., we take a bunch of samples
from all arms at the beginning without adaptively choosing which arm to sample) then
the problem requires $n\log(n)/\Delta^2$ samples.
Therefore, clearly adaptivity helps.

The question the authors study is how this "help" accumulates in time.
Specifically, they look at two key quantities, the mutual information of the best arm
with the history at time $t$ as well as the probability of being able to pick the best arm at time $t$.
They track how these quantities grow as $t$ increases, obtaining a sharp
characterization up to constant factors.
They identify different growth regimes in both cases;
In the case of maximizing probability of identifying the best arm,
the probability is $1/n$ initially (basically random choice)
then linearly increases until it reaches $\Theta(1)$.
In the case of mutual information, the growth consists of 4 phases:
an initial linear growth regime, a quadratic regime, a second linear growth regime and a final
"saturation" regime where the information reaches $\log(n)$ which is the maximum value possible
as it corresponds to the Shannon entropy of the best arm.

**Questions:**

Questions:
- The current bounds seem to assume that the algorithm "plans" for time $t$ from the beginning.
  Do you know how (and if) the results would change if we have a single
  algorithm attempting to give a good answer for all $t$?
  In particular, can an algorithm always obtain the bounds you get
  for all $t$ at the same time?

**Ethical Concerns:**

["NO or VERY MINOR ethics concerns only"]

**Final Justification:**

I have read the author's response. While I see their point about studying simple settings, I had already taken the positive aspects of the clean result in my evaluation. I still have the concern about the narrowness of the current setting and as such keep my score.

**Limitations:**

Yes

**Quality:**

3

**Strengths And Weaknesses:**

Strengths:
- I really like the problem statement as it is very simple
  and intellectually interesting. The results
  are also interesting and paint an interesting picture which was not known before.

Weaknesses:
- While I like the problem, I think it can feel a bit too "toy" for broader interest in the community.
  In particular, the work does not really consider a generic bandit instance
  and while it paints a very detailed picture, the picture is for a single bandit
  instance.

---

> ### Author Rebuttal · Authors · 2025-07-30
>
> We sincerely thank you for your thoughtful feedback and valuable suggestions.
>
> **General response**
>
> We thank the reviewers for suggesting more topics of practical motivation and broader relevance. We will include more discussions accordingly. Indeed, our paper is not really about new algorithms but more a conceptual study about how the information gain (and success probability) behaves in adaptive settings in comparison to non-adaptive ones. The scenario we choose is a natural choice for studying mutual information in adaptive settings with general implications, for the following reasons:
>
> - Upper bounding mutual information is difficult for general adaptive settings and inspires recent works on ridge bandits [1] or linear bandits [2];
>
> - Mutual information is conjectured to be key for understanding sample complexity [2], in addition to recent DEC progresses, which rely on two-point lower bounds; and
>
> - The specific multi-armed bandit case we use is standard for lower bound constructions, reflecting core challenges of the problem. In particular, here the simplicity of this setting is a strength, for it reveals a surprising phenomenon and underscores the need for new analytical tools even in this well-understood baseline scenario.
>
> We make the following contributions by studying this special case:
>
> - We propose simple yet powerful tools to upper bound success probability and mutual information in multi-armed bandits, a canonical problem with a different structure from [1, 2], rendering their techniques inapplicable;
> We reveal a decoupling between mutual information and optimal sample complexity, providing a surprising limitation of the general program in [2].
>
> - We would also like to note that the algorithm is not really our focus: an algorithm which randomly samples $\lceil t\Delta^2 \rceil$ arms and runs a bandit algorithm with optimal regret on them (such as the median elimination algorithm in [3]) would achieve the same success probability. Instead, we use an algorithm that is intuitive (making the source of the logarithmic improvement easy to explain) and simple to analyze the mutual information.
>
> **Specific response**
>
> While I like the problem, I think it can feel a bit too "toy" for broader interest in the community. In particular, the work does not really consider a generic bandit instance and while it paints a very detailed picture, the picture is for a single bandit instance.
>
> - Please refer to our general response. We view the simplicity of this setting as a strength rather than a weakness, for it reveals a surprising phenomenon and underscores the need for new analytical tools even in this well-understood baseline scenario.
>
>
> The current bounds seem to assume that the algorithm "plans" for time t from the beginning. Do you know how (and if) the results would change if we have a single algorithm attempting to give a good answer for all t? In particular, can an algorithm always obtain the bounds you get for all t at the same time?
>
> - Our Algorithm 1 achieves the optimal success probability and mutual information for all $t$ at the same time. If $t$ is not known in advance, we can remove the stopping criterion in Algorithm 1, Line 8, and output $i$ as the current guess for the optimal arm at every step.
>
> [1] Rajaraman, N., Han, Y., Jiao, J., & Ramchandran, K. (2024). Statistical complexity and optimal algorithms for nonlinear ridge bandits. The Annals of Statistics, 52(6), 2557-2582.
>
> [2] Chen, F., Foster, D. J., Han, Y., Qian, J., Rakhlin, A., & Xu, Y. (2024). Assouad, Fano, and Le Cam with Interaction: A Unifying Lower Bound Framework and Characterization for Bandit Learnability. Advances in Neural Information Processing Systems, 37, 75585-75641.
>
> [3] Even-Dar, E., Mannor, S., & Mansour, Y. (2002, June). PAC bounds for multi-armed bandit and Markov decision processes. In International Conference on Computational Learning Theory (pp. 255-270). Berlin, Heidelberg: Springer Berlin Heidelberg.
>
> [4] Garivier, A., Ménard, P., & Stoltz, G. (2019). Explore first, exploit next: The true shape of regret in bandit problems. Mathematics of Operations Research, 44(2), 377-399.

---

> > ### Comment · Reviewer_VwpF · 2025-08-04
> >
> > Thank you for your response.
> > I keep my score; I still think the paper is interesting but as I mentioned in my review, the focus on a special case is in my view an important limitation.

---

### Official Review · Reviewer_NFDf · 2025-07-03

**Clarity:** 1
**Significance:** 2
**Originality:** 3
**Rating:** 3
**Confidence:** 3

**Summary:**

This work studies a stochastic multi-armed bandit instance where one optimal arm outperforms all others by a fixed gap. The authors study the optimal success probability and the mutual information over time. In particular, they show a a linear trend, then a quadratic, then linear again.

**Questions:**

Please try to address my concern 1 above. I'm more than happy to increase my rating as long as clear and convincing answers are provided.

**Ethical Concerns:**

["NO or VERY MINOR ethics concerns only"]

**Quality:**

3

**Strengths And Weaknesses:**

My biggest concerns are the following:

1) I have been trying unsuccessfully to exactly position this work in the literature, which makes it hard to compare their results to existing ones.

There already exist BAI results derived for fixed t:
- Freq. Upper: "Almost Optimal Exploration in Multi-Armed Bandits" by Karnin et al,
- Freq. Lower: "Tight (Lower) Bounds for the Fixed Budget Best Arm Identification
Bandit Problem" by Carpentier and Locatelli,
- Bayesian Upper & Lower: "Bayesian Fixed-Budget Best-Arm Identification" by Atsidakou et al.

Overall, although the authors make a strong case on the importance of studying the different values of the budget $t$, they haven't rigorously stated how existing works (the above and others) perform for different values of $t$.. And if their paper improves upon any existing guarantees.

(Also studying success probability instead of standard error probability (as in the BAI literature) can be misleading in regimes where the success probability is of $\Theta(1)$, but the error is still non-zero)

2) Assuming knowledge of $/Delta$ is usually considered a very limiting practice. It allows to create a priori confidence intervals and test the arms against them.

I am still very skeptical and would love to hear back clarifications from the authors on these.

---

> ### Author Rebuttal · Authors · 2025-07-30
>
> We sincerely thank you for your thoughtful feedback and valuable suggestions.
>
> **General response**
>
> We thank the reviewers for suggesting more topics of practical motivation and broader relevance. We will include more discussions accordingly. Indeed, our paper is not really about new algorithms but more a conceptual study about how the information gain (and success probability) behaves in adaptive settings in comparison to non-adaptive ones. The scenario we choose is a natural choice for studying mutual information in adaptive settings with general implications, for the following reasons:
>
> - Upper bounding mutual information is difficult for general adaptive settings and inspires recent works on ridge bandits [1] or linear bandits [2];
>
> - Mutual information is conjectured to be key for understanding sample complexity [2], in addition to recent DEC progresses, which rely on two-point lower bounds; and
>
> - The specific multi-armed bandit case we use is standard for lower bound constructions, reflecting core challenges of the problem. In particular, here the simplicity of this setting is a strength, for it reveals a surprising phenomenon and underscores the need for new analytical tools even in this well-understood baseline scenario.
>
> We make the following contributions by studying this special case:
>
> - We propose simple yet powerful tools to upper bound success probability and mutual information in multi-armed bandits, a canonical problem with a different structure from [1, 2], rendering their techniques inapplicable;
> We reveal a decoupling between mutual information and optimal sample complexity, providing a surprising limitation of the general program in [2].
>
> - We would also like to note that the algorithm is not really our focus: an algorithm which randomly samples $\lceil t\Delta^2 \rceil$ arms and runs a bandit algorithm with optimal regret on them (such as the median elimination algorithm in [3]) would achieve the same success probability. Instead, we use an algorithm that is intuitive (making the source of the logarithmic improvement easy to explain) and simple to analyze the mutual information.
>
> **Specific response**
>
> Clarification on prior work (see original review)
>
> - Thanks for bringing the references to our attention. We will include a more detailed discussion in the revised version. In short, all the referenced papers provide looser bounds than ours or non-applicable bounds on success probability in the setup we consider. More specifically:
>
>   - Theorem 2 by Carpentier and Locatelli, and Theorem 8 by Atsidakou et al. provide upper bounds on the success probability. They are looser than ours because in our regime, where T \leq C n/\Delta^2, their bounds are at a constant level. A constant level upper bound is vacuous for our regime.
>
>   - Theorem 4.1 by Karnin et al. provides lower bounds on the success probability.  It is also looser than ours because in our regime. Their bound is negative in our regime and hence vacuous.
>
>   - Atsidakou et al. consider the Gaussian case, and the bounds obtained depend on their prior form and are thus not applicable. Meanwhile, Theorems 1 and 2 by Atsidakou et al. provide lower bounds on the success probability, but do not recover the tight linear growth rate compared to ours.
>
> Assuming knowledge of Delta is usually considered a very limiting practice. It allows to create a priori confidence intervals and test the arms against them.
>
> - As in our general response, our focus is a case study about how the information gain (and success probability) could have surprising behaviors in adaptive settings, rather than propose an algorithm for this toy case. We choose to use our random walk algorithm that is intuitive (making the source of the logarithmic improvement easy to explain) and simple to analyze the mutual information.
>
> [1] Rajaraman, N., Han, Y., Jiao, J., & Ramchandran, K. (2024). Statistical complexity and optimal algorithms for nonlinear ridge bandits. The Annals of Statistics, 52(6), 2557-2582.
>
> [2] Chen, F., Foster, D. J., Han, Y., Qian, J., Rakhlin, A., & Xu, Y. (2024). Assouad, Fano, and Le Cam with Interaction: A Unifying Lower Bound Framework and Characterization for Bandit Learnability. Advances in Neural Information Processing Systems, 37, 75585-75641.
>
> [3] Even-Dar, E., Mannor, S., & Mansour, Y. (2002, June). PAC bounds for multi-armed bandit and Markov decision processes. In International Conference on Computational Learning Theory (pp. 255-270). Berlin, Heidelberg: Springer Berlin Heidelberg.
>
> [4] Garivier, A., Ménard, P., & Stoltz, G. (2019). Explore first, exploit next: The true shape of regret in bandit problems. Mathematics of Operations Research, 44(2), 377-399.

---

### Official Review · Reviewer_QCA6 · 2025-07-07

**Clarity:** 3
**Significance:** 4
**Originality:** 4
**Rating:** 5
**Confidence:** 3

**Summary:**

This paper studies the evolution of information in interactive decision making through a toy multi-armed bandit instance where one optimal arm outperforms others by a fixed margin $\Delta$. Providing the related literature and discussing in a detailed fashion, the authors find a good niche that has not yet been analysed and answered. The authors characterize the optimal success probability ($p_t$) and mutual information ($I_t$) over time, revealing distinct growth phases and demonstrating that optimal learning doesn't necessarily require maximizing information gain. The work provides both achievability results and matching converse bounds.

**Questions:**

1. **Extension to other MAB setups**: What are the possible relaxations to the current framework that could potentially be theoretically tractable? What are the barriers preventing extension to the other setups?

2. **Regime Connections**: The authors talk about "true shape of regret". To that, how do the temporal regimes in this paper compare to the regime in the referenced paper? Do they have the same underlying structure or philosophy/intuition?

**Ethical Concerns:**

["NO or VERY MINOR ethics concerns only"]

**Final Justification:**

I am partially satisfied with the author's response on the importance of mutual information. However, I would have liked to see more ramifications of what they prove in the paper regarding the downstream impact.

**Limitations:**

Apart from the toy model for the Multi-armed bandit setup, I do not think there are particular limitations that are not touched upon in the paper.

**Paper Formatting Concerns:**

The paper meets formatting standards, though some notation could benefit from a more careful introduction/definition/intuitive explanation.

**Quality:**

4

**Strengths And Weaknesses:**

**Strengths:**

**Theoretical Sound**: The complete characterization with tight upper and lower bounds across all three regimes is mathematically sound (I have glanced at the proofs; I could have missed some details).

**Surprising Insight**: The analysis of decoupling of $p^*_t$ and mutual information $I^*_t$ is a nice, interesting find and is genuinely surprising and challenges established intuitions about exploration-exploitation trade-offs. This would serve as a nice addition to the fundamental understanding of the multi-armed bandit theory.

**Technical Innovation**: Several proof techniques come together in the proof: the change-of-divergence argument for large t, the reversal of Fano's inequality application, and the careful analysis of temporal transitions. This would serve as a nice template for future research in this direction.

**Literature Integration**: The positioning relative to existing work is comprehensive and demonstrates a clear understanding of the broader landscape.

**Weaknesses:**

**Insufficient Motivation**: The paper fails to adequately justify why mutual information merits study as a fundamental quantity in this context. It would be a great addition to understand the impact of mutual information in popular established algorithms, which would make it a justified problem to pursue.

**Presentation Gaps**: Several technical elements are introduced without a proper foundation (I could be mistaken):
- Equation (2) presents I*_t and I(·,·) without definition
- The "frequentist approach" reference in Remark 1.1 lacks context
- Notation in the change-of-divergence argument (P_H_t, P̄_H_t, P_H_t|a*) appears without explanation

**Further Ramification**: The theoretical insights offer no clear path toward any potential future improvement. After getting a better understanding of the dynamics of interactive decision making, one would assume there are some implications for better algorithmic design to follow.

---

> ### Author Rebuttal · Authors · 2025-07-30
>
> We sincerely thank you for your thoughtful feedback and valuable suggestions.
>
> **General response**
>
> We thank the reviewers for suggesting more topics of practical motivation and broader relevance. We will include more discussions accordingly. Indeed, our paper is not really about new algorithms but more a conceptual study about how the information gain (and success probability) behaves in adaptive settings in comparison to non-adaptive ones. The scenario we choose is a natural choice for studying mutual information in adaptive settings with general implications, for the following reasons:
>
> - Upper bounding mutual information is difficult for general adaptive settings and inspires recent works on ridge bandits [1] or linear bandits [2];
>
> - Mutual information is conjectured to be key for understanding sample complexity [2], in addition to recent DEC progresses, which rely on two-point lower bounds; and
>
> - The specific multi-armed bandit case we use is standard for lower bound constructions, reflecting core challenges of the problem. In particular, here the simplicity of this setting is a strength, for it reveals a surprising phenomenon and underscores the need for new analytical tools even in this well-understood baseline scenario.
>
> We make the following contributions by studying this special case:
>
> - We propose simple yet powerful tools to upper bound success probability and mutual information in multi-armed bandits, a canonical problem with a different structure from [1, 2], rendering their techniques inapplicable;
> We reveal a decoupling between mutual information and optimal sample complexity, providing a surprising limitation of the general program in [2].
>
> - We would also like to note that the algorithm is not really our focus: an algorithm which randomly samples $\lceil t\Delta^2 \rceil$ arms and runs a bandit algorithm with optimal regret on them (such as the median elimination algorithm in [3]) would achieve the same success probability. Instead, we use an algorithm that is intuitive (making the source of the logarithmic improvement easy to explain) and simple to analyze the mutual information.
>
> **Specific response**
>
> Extension to other MAB setups: What are the possible relaxations to the current framework that could potentially be theoretically tractable? What are the barriers preventing extension to the other setups?
>
> - It is shown that the mutual information can be tractable for linear bandits [2]. For ridge bandits, a new notion of information (i.e. the chi-squared informativity) is tractable [1]. So our work complements these in a tabular setting like MAB. The main barrier remains the tractability of mutual information. Indeed, one central contribution of our paper is to provide a novel approach to bounding mutual information.
>
> Regime Connections: The authors talk about "true shape of regret". To that, how do the temporal regimes in this paper compare to the regime in the referenced paper? Do they have the same underlying structure or philosophy/intuition?
>
> - The connection is more on the topic of multi-armed bandits. We focus mainly on the initial phase of learning (where the regret grows linearly in time), a topic which is overlooked in the literature. For example, the “true shape of regret” in [4] treats all regimes in our paper as the linear regret regime, without revealing the surprising growth of success probability and MI.
>
> [1] Rajaraman, N., Han, Y., Jiao, J., & Ramchandran, K. (2024). Statistical complexity and optimal algorithms for nonlinear ridge bandits. The Annals of Statistics, 52(6), 2557-2582.
>
> [2] Chen, F., Foster, D. J., Han, Y., Qian, J., Rakhlin, A., & Xu, Y. (2024). Assouad, Fano, and Le Cam with Interaction: A Unifying Lower Bound Framework and Characterization for Bandit Learnability. Advances in Neural Information Processing Systems, 37, 75585-75641.
>
> [3] Even-Dar, E., Mannor, S., & Mansour, Y. (2002, June). PAC bounds for multi-armed bandit and Markov decision processes. In International Conference on Computational Learning Theory (pp. 255-270). Berlin, Heidelberg: Springer Berlin Heidelberg.
>
> [4] Garivier, A., Ménard, P., & Stoltz, G. (2019). Explore first, exploit next: The true shape of regret in bandit problems. Mathematics of Operations Research, 44(2), 377-399.

---

### Official Review · Reviewer_qePv · 2025-07-07

**Clarity:** 2
**Significance:** 3
**Originality:** 2
**Rating:** 4
**Confidence:** 4

**Summary:**

This paper investigates the mutual information (MI) and the probability-of-error objectives in a classical stationary Bernoulli bandit ranking & selection problem, where the best arm must be identified after a fixed number of samples. The authors provide a rigorous theoretical analysis demonstrating that the policies optimizing cumulative MI about the best arm differ systematically from those optimizing the probability of correctly identifying that best arm. Additionally, the paper shows that standard information-theoretic tools, such as Fano’s inequality, can become loose in certain regimes, motivating novel bounding techniques.

**Questions:**

Can you provide clear, practical scenarios where the MI-based optimization objective would be naturally superior to the classic ranking & selection objective?

Why hasn't the existing ranking & selection literature (knowledge gradient, Bayesian optimization) already made this distinction clear? How does your information-theoretic perspective uniquely illuminate these subtle differences?

Are your insights robust to more general settings (Gaussian or restless bandits)? How would your phase-based analysis of MI accumulation extend to these broader scenarios?

**Ethical Concerns:**

["NO or VERY MINOR ethics concerns only"]

**Final Justification:**

I thank the authors for providing the rebuttal. However, the rebuttal seems to consist largely of identical main text for all reviewers, and in particular do not really address or acknowledge my concerns or suggestions about more concrete discussions of practical utility or better contextualization with other existing work. As such, the rebuttal does not change my mind sufficiently to change my score.

**Limitations:**

No. Actually not at all. I don't see any discussion of limitations, only some propositions in the Discussion. By failing to contextualing the current work conceptually in contrast to other related theoretical work on bandits, the Discussion also missed a chance to say how the current paper doesn't address but can potentially be expanded to address problems such as Gaussian reward distributions, restless bandits, contextual bandits, dueling bandits, etc.

**Paper Formatting Concerns:**

None.

**Quality:**

3

**Strengths And Weaknesses:**

Strengths:

Hidden conceptual depth: While initially appearing somewhat esoteric and purely theoretical, upon careful reflection the paper actually reveals a subtle and important conceptual distinction. Specifically, the authors implicitly characterize conditions under which one should optimize the probability of a correct decision (linear in error probability) versus those conditions under which one should optimize mutual information (logarithmic in error probability). While the former may seem more directly practically useful (choosing the right medication after a trial period, picking the right manufacturing process after prototyping several, etc.), the latter can also be quite useful for practical scenarios involving "communication penalties", or correctional overhead after decisions.

Technical rigor and clarity: The paper presents mathematically rigorous and clearly explained theoretical results. The phase-based analysis of MI accumulation is novel and provides a potentially useful framework for future analyses of related problems.

Potential interdisciplinary relevance: The underlying insights could be relevant to diverse communities (information theory, ranking & selection, reinforcement learning, cognitive modeling), although the authors do not explicitly draw out these connections.

Weaknesses:

Lack of practical framing: A major missed opportunity of this paper is the absence of any clear, practical motivation or interpretation of why and when one would care about the mutual information objective (versus classical success probability). Given the interdisciplinary interest in bandit tasks, it is especially important to provide intuition or motivation that would help a broader audience see practical implications of the theoretical insights. In its way of formulating the problem and relating to existing work, it appeared at first glance a pure intellectual exercise of interest only to information theorist, with little use or interest for the broader NeurIPS community. Upon deeper reflection, I feel that is not actually true (see my first point under Strengths above), but the authors didn't make sufficient attempt to relate to the broader community.

Narrow contextualization: The authors largely frame their work as a novel application of information-theoretic analysis to a standard bandit problem, but they do not sufficiently engage with closely related literature (such as Peter Frazier's "knowledge gradient" and other ranking & selection literature), leaving readers without clear reference points for the importance and novelty of their contribution.

Limited scope: The paper restricts itself to a highly stylized Bernoulli bandit scenario. Demonstrating or discussing generalizations (e.g., structured, contextual or restless bandits) could significantly strengthen the relevance and potential impact of the paper.

I lean toward accepting this paper due to the subtle yet important theoretical insight it provides, which becomes clearer upon deeper reflection. The authors reveal that maximizing probability of correct identification (traditional ranking & selection) and maximizing MI (information-theoretic objective) differ fundamentally. However, it shouldn't be up to the reviewer or reader to have to come up with a motivation for how these two objectives correspond to two different classes of practically meaningful scenarios; it behooves the manuscript to make that front and center from the beginning. Thus, the authors should significantly improve how they communicate the broader practical value and relevance of their findings. Currently, the paper appears narrowly theoretical, diminishing its impact and appeal to the broader NeurIPS community.

---

> ### Author Rebuttal · Authors · 2025-07-30
>
> We sincerely thank you for your thoughtful feedback and valuable suggestions.
>
> **General response**
>
> We thank the reviewers for suggesting more topics of practical motivation and broader relevance. We will include more discussions accordingly. Indeed, our paper is not really about new algorithms but more a conceptual study about how the information gain (and success probability) behaves in adaptive settings in comparison to non-adaptive ones. The scenario we choose is a natural choice for studying mutual information in adaptive settings with general implications, for the following reasons:
>
> - Upper bounding mutual information is difficult for general adaptive settings and inspires recent works on ridge bandits [1] or linear bandits [2];
>
> - Mutual information is conjectured to be key for understanding sample complexity [2], in addition to recent DEC progresses, which rely on two-point lower bounds; and
>
> - The specific multi-armed bandit case we use is standard for lower bound constructions, reflecting core challenges of the problem. In particular, here the simplicity of this setting is a strength, for it reveals a surprising phenomenon and underscores the need for new analytical tools even in this well-understood baseline scenario.
>
> We make the following contributions by studying this special case:
>
> - We propose simple yet powerful tools to upper bound success probability and mutual information in multi-armed bandits, a canonical problem with a different structure from [1, 2], rendering their techniques inapplicable;
> We reveal a decoupling between mutual information and optimal sample complexity, providing a surprising limitation of the general program in [2].
>
> - We would also like to note that the algorithm is not really our focus: an algorithm which randomly samples $\lceil t\Delta^2 \rceil$ arms and runs a bandit algorithm with optimal regret on them (such as the median elimination algorithm in [3]) would achieve the same success probability. Instead, we use an algorithm that is intuitive (making the source of the logarithmic improvement easy to explain) and simple to analyze the mutual information.
>
> **Specific response**
>
> Can you provide clear, practical scenarios where the MI-based optimization objective would be naturally superior to the classic ranking & selection objective?
>
> - The central motivation for studying MI is its universality in establishing lower bounds for regret [2]. More practically, it is possible to come up with bandit scenarios where MI is directly the target, including the well-known information-directed sampling algorithm in the bandit literature, and in cases like measuring a worker’s actual performance in terms of the information acquired as opposed to their final selection.
>
> Why hasn't the existing ranking & selection literature (knowledge gradient, Bayesian optimization) already made this distinction clear? How does your information-theoretic perspective uniquely illuminate these subtle differences?
>
> - Thanks for bringing up these relevant topics. In short, existing works like the knowledge gradient (KG) [Frazier et al.] or Bayesian optimization focus on specific acquisition functions, often tied to expected improvement or posterior variance. While they are related to information acquisition, they do not explicitly study the trajectory of MI. We will include a more detailed discussion in the revised version.
>
> Are your insights robust to more general settings (Gaussian or restless bandits)? How would your phase-based analysis of MI accumulation extend to these broader scenarios?
>
> - Our results and observations still hold for other reward distributions such as Gaussian. Essentially, instead of a biased random walk on integers, we get a biased random walk on real numbers. The same statements will hold for both success probability and mutual information, whereas the analysis gets more complicated. Restless bandits, on the other hand, require a lot more investigation, and we believe it is a valuable future direction.
>
>
> [1] Rajaraman, N., Han, Y., Jiao, J., & Ramchandran, K. (2024). Statistical complexity and optimal algorithms for nonlinear ridge bandits. The Annals of Statistics, 52(6), 2557-2582.
>
> [2] Chen, F., Foster, D. J., Han, Y., Qian, J., Rakhlin, A., & Xu, Y. (2024). Assouad, Fano, and Le Cam with Interaction: A Unifying Lower Bound Framework and Characterization for Bandit Learnability. Advances in Neural Information Processing Systems, 37, 75585-75641.
>
> [3] Even-Dar, E., Mannor, S., & Mansour, Y. (2002, June). PAC bounds for multi-armed bandit and Markov decision processes. In International Conference on Computational Learning Theory (pp. 255-270). Berlin, Heidelberg: Springer Berlin Heidelberg.
>
> [4] Garivier, A., Ménard, P., & Stoltz, G. (2019). Explore first, exploit next: The true shape of regret in bandit problems. Mathematics of Operations Research, 44(2), 377-399.

---

### Decision · Program_Chairs · 2025-09-17

**Decision:**

Accept (poster)

**Comment:**

This paper studies an interesting problem of characterizing the benefits of interactive versus non-interactive decision-making in the context of stochastic bandits. The authors clearly motivate the problem at the beginning using a toy example, and the theoretical results presented are both solid and significant. The reviewers and AC appreciated the contributions of this work. The paper also opens the door for additional research in this direction and is clearly above the acceptance threshold.